# Benchmarking World-Model Learning with Environment-Level Queries

Archana Warrier [1 2]   Thanh Dat Nguyen [1 3]   Michelangelo Naim [1]   Moksh Jain [1 4]   Yichao Liang [1 5]
Karen Schroeder [1]   Cambridge Yang [1]   Joshua B. Tenenbaum [6]   Sebastian Vollmer [2]   Kevin Ellis [7]
Zenna Tavares [1]

## Abstract

World models are central to building AI agents capable of flexible reasoning and planning. Yet current evaluations (i) test only properties measurable from observed interactions, such as next-frame prediction or task return, and (ii) do not test whether a learned model supports diverse queries about the environment. In contrast, humans build *general-purpose* models that can answer many different questions about an environment—including questions that require understanding global structure and counterfactual consequences. We propose *WorldTest*: a protocol for evaluating whether agents learn models that support multiple *environment-level queries*—questions whose answers depend on properties of the full environment, not just observed trajectories. Individually, these queries can target properties (e.g., reachability or the effects of interventions) that no single rollout distribution determines. Collectively, they assess model generality across query types. We instantiate *WorldTest* as *AutumnBench*, a benchmark of 43 interactive grid-world environments and 129 tasks across three query families for both humans and learning agents. Experiments with 517 human participants and five frontier models on AutumnBench show that humans substantially outperform these models, a gap we attribute to differences in exploration and belief updating.

## 1. Introduction

Consider someone who cooks regularly in their own kitchen. Over time, they build an internal model of the workspace—where tools live and how appliances behave—that supports various everyday capabilities. For example, it enables the person to: (1) predict how long the hidden contents of a covered pot will take to finish cooking, based on steam intensity and elapsed time, (2) recognize and adapt to changes in the kitchen (e.g., when knives are placed in a different drawer), and (3) plan a sequence of actions to complete a set of recipes. Cognitive science refers to this flexible, predictive, and counterfactual understanding as *world model* (Weisberg & Gopnik, 2013), a core substrate of human intelligence. Many researchers argue that learning such models is pivotal for the next step in AI progress (LeCun, 2022).

**Benchmarking World-Model Learning with Environment-Level Queries** A core challenge in world-model research is evaluation. Classic reinforcement learning (RL) benchmarks assess agents through task-specific rewards (Brockman et al., 2016). While agents may implicitly build a representation of the environment structure as they pursue rewards—for example, learning that one burner is more efficient because using it leads to faster cooking and thus higher returns—RL benchmarks do not directly test whether the agent learns this structure. To evaluate world models rather than task-specific policies, we need a way to explicitly probe what an agent has inferred about the environment's underlying dynamics.

We call such probes *environment-level queries*: queries whose answers depend on properties of the *entire* environment dynamics, rather than a trajectory of an agent's interaction with the environment. Examples include: (1) inferring what will happen behind an occlusion, (2) detecting changes in the environment's dynamics, and (3) determining whether a state is reachable from another. To assess world-model learning, we should present the query explicitly and treat the agent's response as a direct measure of what it has learned about the environment dynamics.

Existing efforts to evaluate world-model learning, including RL-based benchmarks, only partially assess such capabilities. Cognitive-science-like benchmarks such as Bongard problems and ARC (Depeweg et al., 2018; Chollet, 2019) assess the ability to infer hidden rules from static examples, which are fundamentally environment-level queries about the underlying non-interactive environment. However, they do not test an agent's ability to learn structure through exploration. Conversely, interactive benchmarks such as DiscoveryWorld (Jansen et al., 2024) and CLEVRER vari-

[1]Basis Research Institute [2]DFKI GmbH [3]Harvard University [4]Université de Montréal & Mila - Quebec AI Institute [5]University of Cambridge [6]Massachusetts Institute of Technology [7]Cornell University. Correspondence to: Archana Warrier <archanarw@gmail.com>.

*Proceedings of the 43rd International Conference on Machine Learning*, Seoul, South Korea. PMLR 306, 2026. Copyright 2026 by the author(s).

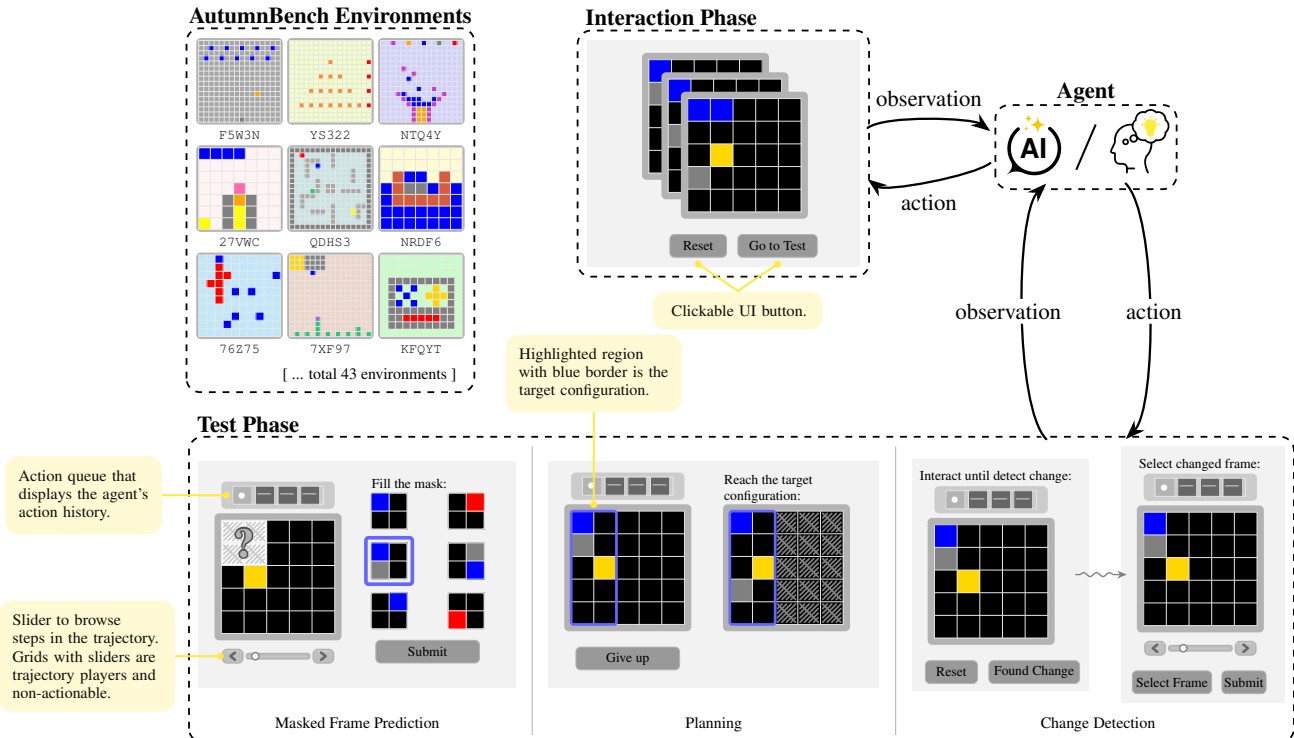

Figure 1. Overview of the WorldTest framework and the AutumnBench instantiation. Agents first interact with an environment without external rewards to build a world model, then act in a derived challenge environment for evaluation. Top-left box shows the 9 example AutumnBench environments. Yellow notes in the middle explain the key UI elements in the human interface of AutumnBench.

ants (Yi et al., 2020) allow exploration but constrain answers to symbolic logics, programs, or natural-language descriptions. Symbolic formats restrict the agent's input–output interface and hinder fair comparison across different agent types, including humans. Natural-language answers are informal and impractical for automated evaluation.

These limitations motivate the central question of this work: *What is an evaluation framework for assessing world-model learning in interactive environments that supports environment-level queries without imposing constraints on the agent's output representations?*

**Our Approach** We introduce *WorldTest*, a behavior-based world-model evaluation framework that probes environment-level queries through a two-phase protocol: interaction and test. In the *interaction phase*, the agent interacts with the environment *without* external rewards. In the *test phase*, WorldTest instantiates an *environment-level query* by transforming the base environment into a *derived challenge environment* with an explicit objective, in which successful completion corresponds to answering the desired query. WorldTest then evaluates the agent based on its ability to achieve this objective using the learned world model.

WorldTest differs from prior evaluation setups in three key ways: (1) **representation-agnostic, automatic scoring**—

WorldTest evaluates agents purely through behavior in the derived challenge environment, without inspecting internal representations or relying on manual or LLM-based judging; (2) **goal-free interaction**—the interaction phase is reward-free, encouraging agents to learn transferable environment dynamics rather than optimize a task-specific objective; and (3) **query-driven generalization**—the test phase compiles environment-level queries into explicit objectives in derived challenge environments, directly measuring whether the learned world model supports downstream tasks and inferences that go beyond the training setting, such as predicting future states, detecting changes in the environment dynamics, and long-horizon planning.

We instantiate WorldTest as a concrete, testable benchmark through *AutumnBench*, consisting of 43 grid-world environments, each paired with three types of test-phase tasks.

The test-phase tasks unify several existing world-model evaluation paradigms such as next-frame prediction and planning by framing them as environment-level queries: (1) *masked-frame prediction (MFP)* where the agent reconstructs masked regions of the final observation from a partially masked trajectory; (2) *change detection (CD)* where the agent detects a change in the environment's dynamics and reports the earliest timestep at which observations diverge from those predicted under the original environment;

and (3) planning where the agent produces an action sequence to reach a target configuration, requiring reasoning about long-term consequences of actions. These three task types capture the capabilities illustrated in the kitchen example above: predicting what's in the covered pot, noticing when drawers are reorganized, and planning recipe steps.

Together, these task-types yield 129 tasks across all environments covering core world-modeling capabilities including prediction, counterfactual reasoning, and planning. While not exhaustive, we designed AutumnBench to be extensible: new environments and challenges can introduce alternative dynamics (e.g., non-standard physics) or evaluate additional skills such as tool use or analogical reasoning.

Finally, we validate AutumnBench through an empirical study involving 517 humans and five state-of-the-art AI models, demonstrating that the benchmark reliably exposes gaps between AI and human world-model learning.

**Contributions** We make the following contributions:

- We propose *WorldTest*, the first theoretical framework for testing world-model learning via *environment-level queries* through an interaction-based evaluation protocol.

- We release *AutumnBench*, a concrete instantiation of WorldTest in a grid-world setting that operationalizes environment-level queries as automated behavioral tests. Additionally, AutumnBench follows the desiderata for novel games outlined by Ying et al. (2025) and is designed for easy extensibility.

- We evaluate human and AI performance on AutumnBench, under the WorldTest framework, demonstrating that AutumnBench reliably exposes gaps between human reasoning and current state-of-the-art models.

In the remainder of the paper, Section 2 reviews related work on addressing the core challenge of environment-level queries, Section 3 provides background on the Autumn language, which we use to define the environments, and the Partially Observed Markov Decision Process (POMDP) formulation of the environments in WorldTest and AutumnBench. Section 4 describes the core WorldTest framework and AutumnBench. Section 5 presents evaluations and analysis of the baseline agents on AutumnBench. Finally, Section 6 concludes and discusses future work.

## 2. Related Works

We categorize approaches to evaluating world-model learning into four non-exclusive categories, detailed in Table 1. Each represents a different and incomplete way of posing environment-level queries that probe the agent's knowledge.

*Non-interactive benchmarks* test if agents can infer rules from a few examples and generalize to novel test cases (Chollet, 2019; Depeweg et al., 2018; Barrett et al., 2018; Kim et al., 2023; Zhang et al., 2020; Yi et al., 2020). While they assess various forms of environment-level reasoning, such as analogy-based concept induction from positive/negative sets of examples (Depeweg et al., 2018), causal reasoning (Zhang et al., 2020; Yi et al., 2020), and rule induction from examples (Chollet, 2019), they do not evaluate learning through interaction with dynamic environments.

*Representation-based approaches* require agents to use predefined output formats—next-frame predictions, programs, or causal graphs—and measure performance via format-specific proxies such as reconstruction error (Srivastava et al., 2015; Ebert et al., 2017), LLM-based evaluation (Jansen et al., 2024), or predicate accuracy (Yi et al., 2020; Girdhar & Ramanan, 2020; Ahmed et al., 2021). These proxies target specific aspects of world understanding but are potentially inadequate, preventing faithful evaluation of learned models or fair comparison against humans.

*Gym-like benchmarks* provide decision-making environments with explicit objectives like rewards in the environment (Brockman et al., 2016; Bellemare et al., 2013; Cobbe et al., 2020; Küttler et al., 2020; Samvelyan et al., 2021; Ahmed et al., 2021; Wang et al., 2021; Tunyasuvunakool et al., 2020). These benchmarks measure task success rather than world-model quality. High performance may arise from memorized policies rather than a generalizable understanding of the environment structure.

*Two-phase RL benchmarks* separate learning from evaluation through two-phase protocols. Unsupervised RL benchmarks (Laskin et al., 2021) let agents explore without objectives, then evaluate on downstream tasks; in-context RL (Laskin et al., 2023) lets agents extract dynamics from interaction history, then act in new tasks. However, both phases occur in the same environment in both settings, and in-context RL interaction histories additionally contain reward signals. Evaluation is thus limited to properties observable through action-reward sequences, precluding tests of the agent's understanding of structural properties, such as regional connectivity or counterfactual variations, that the agent cannot reason about from its trajectories alone.

Existing benchmarks provide diverse, but partial views of world-model learning: non-interactive benchmarks test rule induction from static examples; representation-based approaches constrain outputs to fixed formats; gym-like benchmarks measure task success rather than model quality; and unsupervised RL benchmarks restrict exploration and evaluation to the same environment. None offers a unified mechanism for evaluating environment-level queries that depend on interactive environment dynamics. WorldTest addresses this gap by making such queries central to evaluation, shifting the focus from task performance to whether agents learn how environments work. Crucially, WorldTest is *reward-*

*Table 1.* Comparison of benchmarks in the literature. MDP refers to Markov Decision Process, DET-POMDP refers to Deterministic-Partially Observable Markov Decision Process, and POMDP refers to Partially Observable Markov Decision Process. POMDP is the most general type of environment.

| Benchmark | Environment | Representation Agnostic | Rewardless Training | Modified Test Environment |
|---|---|---|---|---|
| VBench (Huang et al., 2024) | Static | × | ✓ | ✓ |
| SVIB (Kim et al., 2023) | Static | × | ✓ | ✓ |
| CLEVRER (Yi et al., 2020) | Static | × | ✓ | ✓ |
| ACRE (Zhang et al., 2020) | Static | ✓ | ✓ | ✓ |
| RAVEN (Zhang et al., 2019) | Static | ✓ | ✓ | ✓ |
| PGM (Barrett et al., 2018) | Static | ✓ | ✓ | ✓ |
| BONGARD-LOGO (Depeweg et al., 2018) | Static | ✓ | ✓ | ✓ |
| ARC-AGI (Chollet, 2019) | Static | ✓ | ✓ | ✓ |
| PUZZLES (Estermann et al., 2024) | MDP | ✓ | × | ✓ |
| Procgen (Cobbe et al., 2020) | DET-POMDP | ✓ | × | ✓ |
| DiscoveryWorld (Jansen et al., 2024) | POMDP | × | ✓ | ✓ |
| Alchemy (Wang et al., 2021) | POMDP | ✓ | × | × |
| CausalWorld (Ahmed et al., 2021) | POMDP | ✓ | × | × |
| PHYRE (Bakhtin et al., 2019) | POMDP | ✓ | × | ✓ |
| NetHack (Küttler et al., 2020) | POMDP | ✓ | × | × |
| MiniHack (Samvelyan et al., 2021) | POMDP | ✓ | × | × |
| Atari (Bellemare et al., 2013) | POMDP | ✓ | × | × |
| URLB (Laskin et al., 2021) | POMDP | ✓ | ✓ | × |
| **AutumnBench (Ours)** | **POMDP** | ✓ | ✓ | ✓ |

*free* in the interaction phase and *representation-agnostic* in the test phase: rather than requiring agents to produce outputs in a fixed format, it evaluates agents purely through behavior in derived challenge environments, making it possible to assess a broad range of agents, including those whose world knowledge is not expressible in any single predefined representation. These design choices enable evaluations previously inaccessible under reward-centric, representation-constrained, or trajectory-specific approaches.

## 3. Background

In this section, we provide background on the POMDP formulation used by WorldTest to define both the environments and environment-level queries. We also introduce Autumn, the language used to construct AutumnBench environments.

**Partially Observable Markov Decision Processes (POMDPs).** We formulate environments as (reward-free) partially observable Markov decision processes (POMDPs). A POMDP is a tuple $\langle \mathcal{S}, \mathcal{A}, \mathcal{O}, \mathcal{T}, \Omega \rangle$, where $\mathcal{S}$ is the (hidden) state space, $\mathcal{A}$ is the action space, $\mathcal{O}$ is the observation space, $\mathcal{T} : \mathcal{S} \times \mathcal{A} \rightarrow \Delta(\mathcal{S})$ is the transition function that maps state-action pairs to a distribution over next states, and $\Omega : \mathcal{S} \times \mathcal{A} \rightarrow \Delta(\mathcal{O})$ is the observation function that maps state-action pairs to a distribution over observations. Agents interact with the environment by choosing actions and receiving observations at each step, as shown in Figure 1.

**The Autumn Language.** We implement the environments in AutumnBench using the Autumn domain-specific language (DSL). Autumn is a functional reactive language for specifying causal interactions in 2D grids, introduced in

(Das et al., 2023). We chose Autumn as the language for AutumnBench because it allows succinct, expressive specification of the environments, is easy to extend for downstream challenges, and supports both a text-based Gym-like interface for evaluating AI agents and a browser-based graphical user interface for evaluating humans. We provide a detailed description of the Autumn language in Appendix A.

## 4. Theoretical Framework and Benchmark

In the following sections, we first introduce *WorldTest* and then detail its instantiation, *AutumnBench*, a suite of 43 Autumn environments, each with three types of tasks.

### 4.1. WorldTest Framework

We first outline the intuition behind the WorldTest framework and then give a formal definition.

**Intuition.** WorldTest evaluates world-model learning via *environment-level queries*, without constraining the agent's representation or tying evaluation to a fixed reward during interaction. The framework separates interaction from testing and adopts a two-phase protocol: first, a reward-free interaction phase, followed by query-based testing.

1. **Interaction phase**: The agent interacts autonomously with the environment, selecting actions without external goals or rewards. During this phase, it may reset the environment to its initial state as needed, enabling hypothesis testing and systematic exploration. The phase ends when the agent elects to proceed to the next.

2. **Test phase**: The test phase challenges the agent with an *environment-level query*. The protocol encodes this

query as a derived *challenge environment*, obtained by modifying one or more components of the base POMDP (e.g., transitions, observations, or actions), together with an explicit objective. The agent's score depends solely on its behavior in the challenge environment.

We give the formal definition of this framework below.

**Formal Definition.** Let $\mathcal{M} = \langle S, A, O, T, \Omega \rangle \in \mathfrak{M}$ denote the reward-free base environment whose dynamics are unknown to the agent, and let $\mathfrak{M}$ be the space of all possible environments. An *environment-level query* is a computable higher-order function $q : \mathfrak{M} \to \mathcal{Y}$ that maps an environment to an answer in the answer space $\mathcal{Y}$. Unlike queries about individual trajectories, these queries require reasoning about the environment's structure—for example, determining reachability or predicting counterfactual outcomes.

To test whether an agent's world model can answer such queries, WorldTest encodes them as challenge POMDPs with explicit goals. Let $\Xi$ be a query parameter space with associated distribution $P_\Xi$. A WorldTest evaluation protocol specifies $P_\Xi$ and a deterministic function

$$\tau : (\mathcal{M}, \xi) \mapsto (\mathcal{M}', R, H)$$

where $\xi \sim P_\Xi$ is a query instance, $\mathcal{M}' = \langle S', A', O', T', \Omega' \rangle$ is a derived challenge environment, $R : (O' \times A)^H \to \mathbb{R}$ is an objective function, and $H \in \mathbb{N}$ is the evaluation horizon. The challenge environment $\mathcal{M}'$ differs from $\mathcal{M}$ through modifications to the state space, dynamics, observations, or the addition of rewards.

Intuitively, $\tau$ encodes an environment-level query $q$ about $\mathcal{M}$ as a derived challenge environment $\mathcal{M}'$ and objective $R$, such that high scores under $(\mathcal{M}', R, H)$ correspond to correctly answering $q(\mathcal{M})$. Different definitions of $\tau$ yield different task types, such as masked frame prediction, planning, and change detection in AutumnBench.

Given an environment $\mathcal{M}$ and a protocol $(\tau, P_\Xi)$, the events in a concrete evaluation run are as follows:

**Step 1. Disclose task type.** The protocol discloses the task type and structure by describing the transformation function $\tau$, which reveals the challenge action space $A'$, the challenge observation space $O'$, and the form of the reward function $R$ (e.g., binary success or continuous rewards with penalties). The protocol does **not** disclose the task parameters $\Xi$, the modified dynamics $(T', \Omega')$, the explicit reward targets, or any actual observations from $\mathcal{M}'$.

**Step 2. Interact with the environment.** The agent explores the environment $\mathcal{M}$ without external rewards. At any time, the agent may reset the environment to its initial state or proceed to the test phase. During exploration, it collects an interac-

tion history and builds an internal model $\widehat{\mathcal{M}}$ that generalizes to the challenge action and observation spaces $A'$ and $O'$ for the given $\tau$.

**Step 3. Instantiate the protocol.** Once the agent proceeds to the test phase, the protocol samples task parameters $\xi \sim P_\Xi$ and computes $(\mathcal{M}', R, H) = \tau(\mathcal{M}, \xi)$.

**Step 4. Produce a policy from the internal model.** Using its internal model $\widehat{\mathcal{M}}$ and knowledge of $\tau$, the agent returns a policy $\pi : (O')^* \to \Delta(A')$.

**Step 5. Execute the policy in the challenge environment.** The agent runs $\pi$ in $\mathcal{M}'$ for $H$ steps, obtaining a history $h' = (o'_0, a'_0, \ldots, o'_H, a'_H)$.

**Step 6. Score.** The protocol computes the score $R(h')$.

The interaction phase imposes no time limit, but agents only receive scores if they proceed to and complete the test phase. While, in theory, an agent might interact indefinitely, WorldTest conditions evaluation on agents eventually moving on to testing. In practice, however, finite resources prevent indefinite exploration.

### 4.2. AutumnBench: Design and Implementation

In this section, we present an instantiation of the WorldTest framework, namely, AutumnBench. We instantiate WorldTest by implementing three $\tau$ functions for the three task-types: masked frame prediction ($\tau_{\text{MFP}}$), change detection ($\tau_{\text{CD}}$), and planning ($\tau_{\text{PL}}$). AutumnBench consists of 129 *problems*. Each *problem* in AutumnBench has two parts: (1) a base environment with no external rewards that the agent interacts with in the interaction phase, and (2) a derived challenge environment for the test phase.

We describe each of those parts in more detail below, beginning with the environments used in the interaction phase and followed by the related challenge tasks for the test phase:

**Interaction-Phase Environments.** Each AutumnBench environment is a 2D grid world composed of objects, represented as collections of pixels, along with their dynamics. We define these objects and dynamics programmatically using the Autumn language (Das et al., 2023).

The grid sizes of these environments range from $3 \times 3$ to $25 \times 25$, with most at $16 \times 16$. Each environment has five or fewer object types and 1 to 12 colors, with 17 of the 43 environments being stochastic: the same action taken from the same state can produce different outcomes due to randomness in the environment's transition dynamics. Table B.2 summarizes complexity metrics across all environments. Our environments meet the three *novel game* desiderata from Ying et al. (2025) for evaluating world model learning: they are structurally novel, intuitive to humans, and diverse in both world dynamics and learning mechanisms. Figure 1

shows some example environments. AutumnBench includes Atari-like games (Bellemare et al., 2013), simulations of real-world phenomena like plant growth and sandcastle construction, and strategic games like Nim (Wikipedia, 2025). Appendix B gives further details on the different classes of environments in AutumnBench. The action space in Autumn is: `left`, `right`, `up`, `down`, clicking on a grid cell, and `no-op`, which advances the environment by one timestep with no user input. The interaction phase adds a `reset` action that restores the initial state.

**Test-Phase Challenges.** We selected the test-phase challenges to address various aspects of existing world model evaluations. Specifically, AutumnBench implements the test phase of WorldTest by assessing the agent through the following three types of challenges:

- **Masked Frame Prediction (MFP)**: The agent observes a trajectory with partially masked frames and predicts the missing content in the final frame by selecting from six options, only one of which matches the ground truth. The agent scores 1 for a correct selection and 0 otherwise.

- **Change Detection (CD)**: The agent interacts with a modified version of the base environment in which one of the rules changes at some point during the test-phase interaction. It must identify the earliest timestep at which the rule changes. The agent scores 1 for detecting the change at the correct timestep, partial credit for late detection, and 0 for selecting a timestep before the change.

- **Planning**: The agent receives a goal in the form of a target configuration for a subgrid of the environment. It interacts with the base environment, in which the `reset` action is no longer available, and must produce a sequence of actions that drives the subgrid into the target configuration. The agent scores 1 for reaching the goal and 0 otherwise.

In AutumnBench, we implement Step 1 of the WorldTest protocol in Section 4.1 using interface-specific means: interactive tutorials for humans and text descriptions for reasoning models, further described in Section 5. However, the agents do not learn the specific task parameters $\xi$ at this stage, such as which frames are masked, when the change occurs, or which goal states are designated.

We provide the full formal definitions of each challenge and a complete specification of the environments in Appendix C.

# 5. Baseline Evaluations and Analysis

We evaluate humans and reasoning models on Autumn-Bench using the two-phase WorldTest framework. We recruit human participants via Prolific (Prolific, 2025). For model baselines, we use Claude 4 Sonnet, Gemini 2.5 Pro and Flash, o3, and Qwen3 235b a22b thinking 2507. Section 5.1 details our evaluation setup for both. We addi-

tionally include an Autumn-simulator baseline agent with access to the ground-truth Autumn programs, which serves as a reference point for next-frame prediction style world models; see Section E.2.1 for details.

In Section 5.2, we analyze agent performance, their exploration strategies during the interaction phase, and how additional computational resources affect their scores across AutumnBench tasks. We then discuss what these findings reveal about current reasoning models in Section 5.3.

## 5.1. Evaluation Setup

Here, we describe how we tested agents on AutumnBench, including participant selection and filtering criteria, and the implementation details of the interface for each agent.

### 5.1.1. HUMAN EVALUATION

We recruited 517 English-speaking participants via Prolific, screening out color-blind participants and those who failed attention and comprehension checks (Muszyński, 2023). We repeated each problem 20 times, distributing them uniformly so each participant solved five problems of the same type.

Crowd-sourced responses exhibited high variability despite screening (Reid et al., 2022; Douglas et al., 2023), often reflecting differences in effort rather than cognitive capability. To measure performance of a baseline agent representing the average engaged human, we constructed an aggregate agent using the 80th percentile score per problem across the 20 attempts. Throughout our analysis, "human" performance refers to this single 80th-percentile aggregate agent.

We implemented AutumnBench using a web-based graphical user interface (GUI). Participants receive a tutorial on the GUI for their assigned task type, describing the query function $\tau$ being used ($\tau_{\text{MFP}}$, $\tau_{\text{CD}}$, or $\tau_{\text{Planning}}$), per Step 1 of the WorldTest protocol in Section 4.1. Figure 1 shows a rendered grid and the buttons for `reset` and `go-to-test` actions during the interaction phase. The GUI updates at a fixed environment-dependent frame rate of 3–8 FPS, creating real-time exploration experiences. Participants use directional arrow keys or click grid cells at each timestep; inaction triggers automatic `no-op` updates. Section D.2.1 details the GUI implementation for each task type.

### 5.1.2. REASONING-MODEL EVALUATION

We evaluated five frontier reasoning models—Claude 4 Sonnet, Gemini 2.5 Pro and Flash, o3, and Qwen3-235b-a22b-thinking-2507—on AutumnBench. We gave one Autumn-Bench problem to each model at a time. Due to API cost constraints, we evaluated each model's performance based on a single trajectory completion per problem.

At each timestep, we give each model its full interaction history, current grid state, available actions, and a descrip-

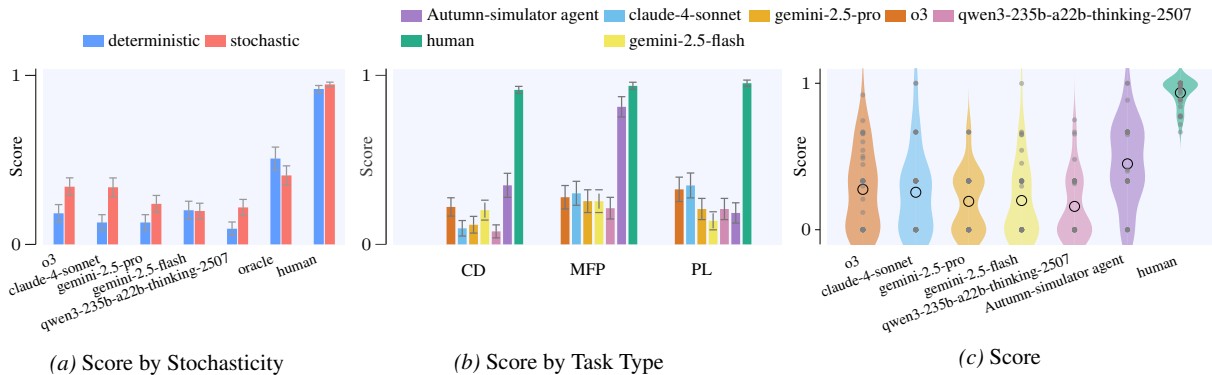

*(a) Score by Stochasticity*    *(b) Score by Task Type*    *(c) Score*

*Figure 3.* Aggregate scores over all AutumnBench problems. (a) Reasoning models perform better in stochastic environments than in deterministic ones, while humans perform consistently across both. (b) Humans outperform reasoning models across the three task families: CD, MFP, and planning. (c) Environment-wise score distributions per agent across the 129 AutumnBench problems.

tion of the task type (i.e., the $\tau$ function), per Step 1 of the WorldTest protocol in Section 4.1. We represent grid states as two-dimensional arrays of color strings; see Figure 2.

```
[
    ["blue",  "black",  "black", "black", "black"],
    ["grey",  "black",  "black", "black", "black"],
    ["black", "yellow", "black", "black", "black"],
    ["black", "black",  "black", "black", "black"],
    ["black", "black",  "black", "black", "black"]
]
```

*Figure 2.* Textual representation of the top grid in the "interaction phase" of Figure 1.

While humans face an implicit time limit from the fixed frame rate, reasoning models step through the environment untimed. Models can also choose not to take any action via a `no-op`. As with humans, during the interaction phase models can select the `go-to-test` action to enter the test phase or the `reset` action to restore the initial state. Section D.2.2 gives implementation details per task type.

### 5.2. Results

In this section, we examine three aspects of agent behavior on AutumnBench: task performance, exploration patterns, and the focusing of behavior during world-model learning.

#### 5.2.1. OVERALL PERFORMANCE

Humans outperformed all five reasoning models across every environment and task type; see Figure 3b. Table F.2 reports detailed scores per agent and problem, and Figure 3c shows environment-wise score distributions.

To understand what drives reasoning-model performance, we first examined how environmental stochasticity affects different agents. Humans performed nearly identically in both deterministic and stochastic environments. Reasoning models, however, did significantly better in stochastic

settings than in deterministic ones; see Figure 3a.

Next, we examined how computational cost affects model performance. Ranking the five reasoning models by ascending cost-per-problem—Gemini 2.5 Flash, Qwen3-235B, Gemini 2.5 Pro, Claude 4 Sonnet, o3—we split the 43 environments by whether average score increased monotonically along this ordering. Figure 4 shows average score against cost-per-problem for each agent, where the environments are split into: `SetA`, with 25 environments, where performance improved monotonically with cost, and `SetB`, with 18 environments, where additional compute gave no benefit.

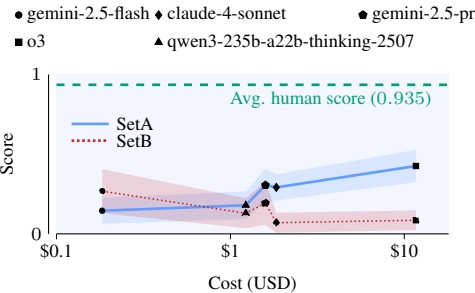

*Figure 4.* Score vs. cost per problem across environments.

Within each task type, higher cost improved performance in 37% of masked-frame prediction and planning tasks, and 33% for change detection; see Section F.1.3 for the detailed breakdown. That 42% of environments do not benefit from additional compute suggests reasoning limitations beyond scaling. Appendix F.1 gives environment-specific results.

#### 5.2.2. EXPLORATION PATTERNS

We analyze agents' actions during exploration using two complementary methods. First, we analyze the distribution of unique actions by counting each distinct click position and directional action. For example, clicking on positions $(2, 3)$ and $(5, 7)$ counts as two unique click actions. This

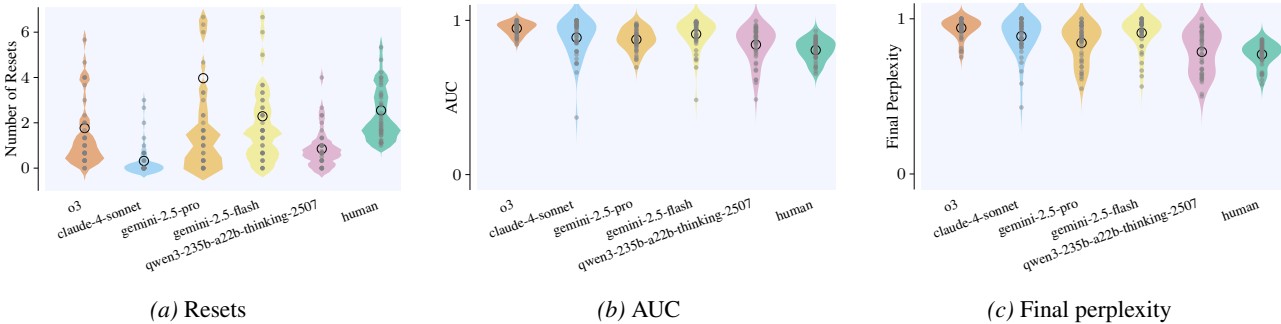

*(a)* Resets        *(b)* AUC        *(c)* Final perplexity

*Figure 6.* Environment-specific performance and behavioral metrics. (a) Reset frequency distributions, (b) Area Under the Curve (AUC) for normalized perplexities, and (c) Final perplexity values.

approach allows us to account for environment-specific variations within AutumnBench, such as changes in static object positions and grid sizes. We then calculate the fraction of unique actions in each category—clicks, directional actions, `reset`, and `no-op`—relative to the total number of unique actions the agent took, as shown in Figure 5.

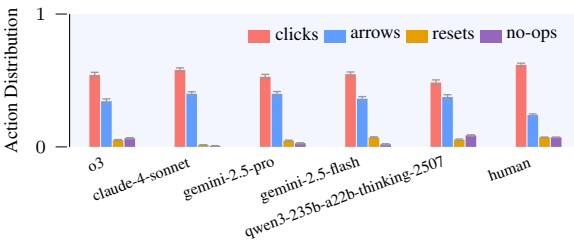

*Figure 5.* Action type distribution per agent averaged over AutumnBench problems.

Humans use roughly equal proportions of `reset`s and `no-op`s at 12.5% each, while reasoning models focus on clicks and directional actions. Every reasoning model spends less than 7% of its actions on `reset`s and `no-op`s combined, with Claude the lowest at 2.1%.

Second, we examine reset frequencies in Figure 6a. Humans reset at least once in every environment and exceed the per-environment model mean in 33 of 43. Models frequently skip resets entirely: Claude in 31 environments, Qwen in 8, Gemini Pro in 7, Gemini Flash in 4, and o3 in 3.

Both analyses suggest that reasoning models do not treat resets as special actions, unlike humans. Table F.3 reports average actions per environment for humans and models.

This difference is not only quantitative but also structural: around resets, humans tend to replay similar action subsequences, yielding a much higher longest common subsequence (LCS) ratio between pre- and post-reset action windows than reasoning models (human mean 0.637 vs. 0.188–0.360 for models), consistent with more systematic hypothesis testing; Section F.2.3 reports the full analysis.

### 5.2.3. WORLD-MODEL LEARNING

We quantify how agents acquire world models by measuring how their actions become more focused over time. Specifically, we use *normalized perplexity*, defined in Definition D.1, to quantify how predictable an agent's actions are while exploring. High perplexity suggests random actions, while low perplexity suggests more targeted behavior.

We analyze normalized perplexity across the full interaction using two metrics. The Area Under the Curve (AUC) of normalized perplexity, defined in Definition D.2, captures the overall learning trajectory. Agents that quickly develop focused exploration strategies show lower AUC values. Final perplexity measures how targeted agents become by the end of exploration. Both metrics are robust to brief fluctuations and enable fair comparison across environments and agents.

Humans show lower values on both measures, as seen in Figures 6b and 6c, suggesting more effective world-model learning. Lower AUC values indicate a rapid transition from random clicks and keypresses to more targeted actions. Their final normalized perplexity values fall consistently below those of reasoning models, suggesting more deterministic and purposeful behavior by the end of the exploration. Figure 6 visualizes these trends across all environments.

### 5.3. Discussion

Our results reveal differences in how humans and reasoning models approach world-model learning along two axes: experimental design and belief updating. These differences reflect limitations in their ability to monitor and update their learning processes, including decisions about what information to acquire and when to revise their beliefs.

**Experimental design.** We conjecture that humans use resets as experimental tools to test hypotheses about environmental dynamics. Reasoning models also form and test hypotheses, but their reasoning traces suggest a narrow view of what counts as informative actions, as illustrated in Listing F.1: they prioritize keypresses and clicks, fail-

ing to recognize that `resets` and `no-ops` can be equally valuable for hypothesis testing. This is consistent with the action distributions reported above in Figure 5 and with the reset-structure analysis above.

**Belief updating.** Reasoning models often fail to update their understanding when faced with contradictory evidence, especially in masked-frame prediction tasks as seen in Listing F.1. Even when they recognize that test-phase observations contradict the rules learned during interaction, they tend to rely on those original rules in their predictions.

Collectively, these results suggest that the human–model gap is associated with limitations at multiple levels of inference. Achieving human-level performance likely requires not only better priors over world models and memory, but also advances in strategic experimental design, uncertainty quantification, and flexible belief updating.

## 6. Conclusion

In this work, we present WorldTest, a behavior-based framework that evaluates world-model learning through environment-level queries: derived challenge environments that agents answer by acting in them. Unlike prior approaches that constrain agent representations or evaluate only through reward optimization, WorldTest uses a two-phase protocol: reward-free interaction followed by scored evaluation in these challenge environments. Scoring purely on behavior lets WorldTest measure whether learned world models support downstream reasoning while enabling fair comparison across diverse agent types, including humans.

We instantiate WorldTest in AutumnBench, which consists of 43 interactive grid-world environments and 129 tasks across masked-frame prediction, change detection, and planning. Our experiments with 517 human participants and five frontier reasoning models on AutumnBench reveal substantial gaps between human and AI world-model-learning capabilities. Humans outperform these models across all environments and task types, achieving near-optimal scores while the models frequently fail. We also find that humans devote a higher fraction of their actions to resets than the models do and achieve lower perplexity over their trajectories. This association is consistent with the hypothesis that humans use resets as an exploration or hypothesis-testing mechanism, but establishing a causal link requires interventional designs that we leave to future work. Current frontier models lack the flexible, predictive understanding that characterizes human-like world models.

While AutumnBench demonstrates WorldTest in a grid-world setting, the framework is broader: future work may instantiate WorldTest in physics-rich environments, robotics, multi-agent systems, and other domains with rich dynamics. Model performance is sensitive to prompt formulation,

as shown in Section F.1.2, though the human–model gap remains large across all variants tested. We recommend that future evaluations report results across multiple prompt formulations with variance bars.

## Code and Data Availability

Warrier et al. (2026) provides the public AutumnBench dataset. To protect benchmark integrity and prevent data leakage, we keep a subset of problems private but provide full specifications for all reported experiments. The web-based GUI is at autumn.basis.ai, the Autumn interpreter (with WASM, Python, and Julia bindings) is on GitHub as BasisResearch/Autumn.cpp, and the reasoning-model baselines are at BasisResearch/MARAProtocol.

## Author Contributions

Archana Warrier led the project and designed AutumnBench, implementing the Autumn programs with Dat Nguyen and designing the downstream evaluation challenges. She also collected human and model baselines and conducted the statistical analysis. Dat Nguyen built the Autumn interpreter and the agent-protocol codebase. Dat Nguyen, Michelangelo Naim, Cambridge Yang, and Archana Warrier built the website and human-facing UI. Moksh Jain developed the reasoning-model baseline infrastructure. Yichao Liang worked on Autumn-simulator agent baselines and the protocol. Cambridge Yang served as primary advisor, and Zenna Tavares as senior advisor. All authors contributed to conceptualization, writing, and revision.

## Acknowledgments

We thank Ria Das for contributions to the original Autumn project; Julian Jara-Ettinger and Tobias Gerstenberg for feedback on the experimental design; Yixiu Zhao, Evan Pu, and Emily Mackevicius for manuscript review; and Benjamin Crouzier and Tufa Labs for support and feedback. Moksh Jain is supported by a FRQNT Doctoral Fellowship (doi:10.69777/366694).

## Impact Statement

This paper presents work whose goal is to advance the field of Machine Learning. There are many potential societal consequences of our work, none which we feel must be specifically highlighted here.

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

# Appendices

## A. The Autumn Language

We implement the POMDP environments in AutumnBench using the Autumn domain-specific language (DSL). Autumn is a functional reactive language for specifying causal interactions in 2D grids, introduced in (Das et al., 2023). We chose Autumn as the base language for AutumnBench because it allows for succinct and expressive specification of environments, is easily extensible to implement downstream challenges, and supports both a text-based Gym-like interface for evaluating AI agents and a browser-based graphical user interface for evaluating human participants.

An Autumn program has the following parts:

- *Environment setup* defines the grid size ($n \times n$) and background color.

- *Object type definitions* specify each object type by listing its shape (as relative 2D coordinates and colors) and its internal fields. These fields record state variables unique to each object instance.

- *Object instance definitions* define the initial state of each object instance and the default function for each object instance under the `no-op` action.

- *Event handlers* establish the rest of the full transition function $T$. In particular, an `on` clause pairs a predicate $p(s, a)$ with an intervention $i$, so that if $p(s, a)$ holds, the state transitions to $s' = i(s)$.

An Autumn program defines a POMDP. Specifically, the global variables in the program correspond to the hidden state. The agent observes only the grid's color matrix ($n \times n$), where every cell occupied by the highest-z-order object instance is visible; all other state components remain latent. The action space of an Autumn environment includes: 1. the `no-op` action, 2. four directional actions—up, down, left, right—corresponding to the four arrow keys on a typical keyboard, and 3. a family of click actions click$(x, y)$, corresponding to clicking on the cell at position $(x, y)$ with a pointer device.

**Example Autumn Environment.** To illustrate the Autumn language, we present a treasure-hunting environment in which an agent searches for hidden treasures using a noisy distance sensor. Listing A.1 shows the complete Autumn program implementing this environment, and Figure A.1 shows a rendered trajectory sampled from it. At the start, the environment reveals only one treasure—the gold grid cell in the middle of the grid. The agent controls a sensor probe, the grey grid cell, and moves it using four directional actions: up, down, left, and right. The top blue horizontal bar displays the probe's noisy estimate of its distance to the nearest treasure. If the agent clicks a hidden treasure, the environment reveals it.

```
1  (= GRID_SIZE 5) (= BACKGROUND "black")          ← Environment setup
2
3  (object Proximity (dist)
4    (map (--> (i) (Cell i 0 (if (<= i dist) "blue" "black" )))
5      (range 0 GRID_SIZE)))
6  (object Treasure (revealed) (Cell 0 0 (if revealed "gold" "black")))   ← Object definitions
7  (object Agent (Cell 0 0 "grey"))
8
9  (= t1 (initnext (Treasure true (randomPosition 0 1 GRID_SIZE GRID_SIZE)) (prev t1)))
10 (= t2 (initnext (Treasure true (randomPosition 0 1 GRID_SIZE GRID_SIZE)) (prev t2)))
11 (= t3 (initnext (Treasure false (randomPosition 0 1 GRID_SIZE GRID_SIZE)) (prev t3)))   ← Object instances;
12 (= agent (initnext (Agent (Position 0 1)) (prev agent)))                they define the initial state
13 (= tdist (fn (treasure) (sqdist (.. agent origin) (.. treasure origin))))   and the default transitions
14 (= proximity (initnext (Proximity 2 (Position 0 0)) (updateObj proximity "dist"   with the no-op action
15      (+ (min (min (tdist t1) (tdist t2)) (tdist t3))
16      (- (uniformChoice (range 0 3)) 1)))))
17
18 (on up (= agent (moveUp agent)))
19 (on down (= agent (moveDown agent)))
20 (on left (= agent (moveLeft agent)))
21 (on right (= agent (moveRight agent)))                          ← Event handlers ("on" clauses)
22 (on (clicked t1) (= t1 (updateObj t1 "revealed" true)))
23 (on (clicked t2) (= t2 (updateObj t2 "revealed" true)))
24 (on (clicked t3) (= t3 (updateObj t3 "revealed" true)))
```

*Listing A.1.* A treasure-hunting environment example in Autumn. Interactive demo available at autumn.basis.ai.

*Figure A.1.* Renderings of a partial trajectory of the Autumn environment example from Listing A.1, following the sequence of actions down, down, down and click(1, 4).

## B. AutumnBench Environment Details

AutumnBench's 43 environments span several axes: what drives the dynamics, what reasoning the task demands, and how the agent interacts with the grid. We tag each environment with one or more of the nine non-exclusive labels below; Table B.1 gives the full assignment, and Table B.2 reports program-level statistics.

**Board & Puzzle Games (14 envs)** The environment mirrors a known turn-based or board/puzzle game (e.g., YS322, 27VWC). *Table tag:* Games.

**Atari-Style (6 envs)** The agent drives a single avatar through an environment resembling an Atari game (e.g., F5W3N). *Table tag:* Atari.

**Multi-step Reasoning (7 envs)** Solving the task requires chaining several steps of reasoning—either because the environment exposes multiple controllable units whose per-unit capabilities and interactions must be coordinated, or because the agent must deduce a rule through a sequence of dependent inferences (e.g., 7XF97, QDHS3). *Table tag:* Multi-step.

**Emergent Dynamics (4 envs)** Global structure arises from simple local rules applied uniformly across the grid. Patterns and aggregates are not encoded directly but emerge from many local interactions (e.g., 76Z75). *Table tag:* Emergent.

**Physical Analogue (15 envs)** The mechanics map onto an intuitive real-world phenomenon such as buoyancy, plant growth, or building sandcastles (e.g., 7XF97, NRDF6). *Table tag:* Physical.

**Visible Autonomous Dynamics (25 envs)** The world evolves visibly between agent actions, even under no-op; the agent is not the only driver of change (e.g., 76Z75, NTQ4Y). *Table tag:* Visible Auto. Dyn.

**Latent Autonomous Dynamics (2 envs)** Hidden state evolves on its own, even under no-ops, without the agent seeing the progression directly—for example, a hidden timer (27VWC). *Table tag:* Latent Auto. Dyn.

**Boolean Logic (3 envs)** The agent must identify a Boolean rule governing some latent states of objects (e.g., KFQYT). *Table tag:* Logic.

**Clickable Cells (11 envs)** Static cells act as buttons that the agent clicks to trigger state changes, alongside any other available interactions (e.g., NTQ4Y, YS322). *Table tag:* Clickable Cells.

*Table B.1.* Category tags for each AutumnBench environment. Labels are non-exclusive.

| # | Env | Categories | # | Env | Categories |
|---|-----|------------|---|-----|------------|
| 1 | S2KT7 | Visible Auto. Dyn. | 23 | 4CKC2 | Logic, Physical, Clickable Cells |
| 2 | 27VWC | Latent Auto. Dyn., Clickable Cells, Games | 24 | 7WWW9 | Physical |
| 3 | KFQYT | Logic | 25 | N2NTD | Atari, Visible Auto. Dyn. |
| 4 | B58F3 | Atari, Visible Auto. Dyn. | 26 | BY2Q7 | Games, Clickable Cells |
| 5 | NRDF6 | Physical, Visible Auto. Dyn. | 27 | 4T8TR | Games, Visible Auto. Dyn. |
| 6 | 6JKKA | Atari, Visible Auto. Dyn. | 28 | YS322 | Games, Clickable Cells |
| 7 | K8MTQ | Games, Multi-step | 29 | WHGHP | Atari, Visible Auto. Dyn. |
| 8 | T5F9B | Games | 30 | EAHCW | Games |
| 9 | N59TE | Games, Visible Auto. Dyn. | 31 | 83WKQ | Visible Auto. Dyn. |
| 10 | 76Z75 | Visible Auto. Dyn., Emergent | 32 | 3J4Z7 | Games, Multi-step |
| 11 | NF5VZ | Visible Auto. Dyn., Physical, Emergent | 33 | QDHS3 | Games, Visible Auto. Dyn., Multi-step |
| 12 | 236VK | Atari, Visible Auto. Dyn. | 34 | VA6FQ | Visible Auto. Dyn., Physical, Clickable Cells, Multi-step, Emergent |
| 13 | DQ8GC | Visible Auto. Dyn., Multi-step | 35 | 7VKTD | Games, Visible Auto. Dyn. |
| 14 | QM9XB | Physical, Clickable Cells | 36 | VZ2Q4 | Games, Visible Auto. Dyn. |
| 15 | 6JVMF | Visible Auto. Dyn. | 37 | XHGKQ | Games, Visible Auto. Dyn. |
| 16 | 27JBD | Visible Auto. Dyn., Clickable Cells, Emergent | 38 | F5W3N | Atari, Visible Auto. Dyn. |
| 17 | VQJH6 | Visible Auto. Dyn., Physical, Clickable Cells | 39 | JXQAW | Games |
| 18 | QQM74 | Visible Auto. Dyn., Physical | 40 | 4N7BB | Games |
| 19 | 7XF97 | Visible Auto. Dyn., Physical, Multi-step | 41 | NTQ4Y | Physical, Visible Auto. Dyn., Clickable Cells, Multi-step |
| 20 | BT3GB | Visible Auto. Dyn., Physical, Clickable Cells | 42 | DGG2C | Physical |
| 21 | BT2KZ | Physical | 43 | B8AKZ | Latent Auto. Dyn., Physical |
| 22 | 9F8AJ | Logic, Clickable Cells, Physical | | | |

*Table B.2.* Program statistics for all AutumnBench environments. Columns: number of object types (#Obj), number of latent variables (#Lat., i.e. Autumn program fields not rendered on the grid), program length in lines of code (Len.), number of event handlers (#On), number of colors (#Col.), and grid size. **Left:** deterministic environments. **Right:** stochastic environments.

| | Deterministic | | | | | | | | Stochastic | | | | | |
|---|-----|------|------|-----|------|------|---|-----|------|------|------|-----|------|------|
| # | Env | #Obj | #Lat. | Len. | #On | #Col. | Grid | # | Env | #Obj | #Lat. | Len. | #On | #Col. | Grid |
| 1 | 27VWC | 4 | 4 | 19 | 4 | 10 | 7 | 27 | VZ2Q4 | 1 | 4 | 18 | 3 | 12 | 20 |
| 2 | KFQYT | 2 | 5 | 21 | 8 | 6 | 11 | 28 | S2KT7 | 2 | 0 | 10 | 3 | 2 | 16 |
| 3 | NRDF6 | 3 | 1 | 25 | 1 | 3 | 7 | 29 | B58F3 | 3 | 2 | 22 | 2 | 5 | 13 |
| 4 | 6JKKA | 5 | 8 | 54 | 6 | 6 | 13 | 30 | N59TE | 1 | 4 | 21 | 3 | 5 | 10 |
| 5 | K8MTQ | 2 | 2 | 32 | 3 | 3 | 8 | 31 | 76Z75 | 1 | 1 | 8 | 1 | 2 | 10 |
| 6 | T5F9B | 5 | 6 | 28 | 6 | 8 | 7 | 32 | NF5VZ | 3 | 1 | 18 | 3 | 3 | 9 |
| 7 | DQ8GC | 1 | 1 | 14 | 6 | 2 | 16 | 33 | 236VK | 3 | 1 | 18 | 6 | 4 | 20 |
| 8 | QM9XB | 3 | 3 | 24 | 7 | 5 | 16 | 34 | 6JVMF | 1 | 0 | 7 | 1 | 1 | 16 |
| 9 | 27JBD | 2 | 2 | 17 | 3 | 4 | 16 | 35 | BT2KZ | 1 | 2 | 34 | 1 | 5 | 25 |
| 10 | VQJH6 | 2 | 2 | 24 | 9 | 5 | 17 | 36 | BY2Q7 | 4 | 4 | 52 | 2 | 12 | 12 |
| 11 | QQM74 | 2 | 3 | 16 | 6 | 2 | 21 | 37 | 4T8TR | 2 | 3 | 8 | 1 | 3 | 10 |
| 12 | 7XF97 | 4 | 2 | 24 | 8 | 5 | 16 | 38 | WHGHP | 4 | 4 | 42 | 6 | 6 | 10 |
| 13 | BT3GB | 3 | 2 | 18 | 4 | 4 | 16 | 39 | 83WKQ | 1 | 0 | 5 | 1 | 1 | 16 |
| 14 | B8AKZ | 2 | 3 | 15 | 3 | 5 | 9 | 40 | 3J4Z7 | 1 | 4 | 17 | 1 | 4 | 5 |
| 15 | 9F8AJ | 4 | 3 | 34 | 6 | 8 | 24 | 41 | XHGKQ | 2 | 1 | 24 | 4 | 3 | 16 |
| 16 | 4CKC2 | 3 | 4 | 36 | 2 | 6 | 24 | 42 | F5W3N | 4 | 2 | 35 | 14 | 5 | 16 |
| 17 | 7WWW9 | 1 | 1 | 22 | 8 | 2 | 16 | 43 | 4N7BB | 1 | 2 | 38 | 5 | 9 | 3 |
| 18 | N2NTD | 5 | 5 | 34 | 10 | 6 | 12 | | | | | | | | |
| 19 | YS322 | 2 | 1 | 27 | 8 | 3 | 17 | | | | | | | | |
| 20 | EAHCW | 1 | 3 | 17 | 9 | 5 | 16 | | | | | | | | |
| 21 | QDHS3 | 4 | 6 | 141 | 36 | 6 | 24 | | | | | | | | |
| 22 | VA6FQ | 3 | 3 | 23 | 5 | 5 | 10 | | | | | | | | |
| 23 | 7VKTD | 3 | 2 | 57 | 5 | 3 | 20 | | | | | | | | |
| 24 | JXQAW | 1 | 3 | 38 | 2 | 4 | 5 | | | | | | | | |
| 25 | NTQ4Y | 4 | 2 | 34 | 9 | 5 | 16 | | | | | | | | |
| 26 | DGG2C | 2 | 2 | 18 | 6 | 2 | 17 | | | | | | | | |

# C. AutumnBench Challenge Formulation

This section provides the formal definitions of the three task types in AutumnBench: masked frame prediction, change detection, and planning. For each task type, we define the task structure and scoring function.

## C.1. Masked Frame Prediction

We define each masked frame prediction task by a tuple of a fixed action sequence $\mathbf{a} = (a_1, a_2, \ldots, a_T) \in \mathcal{A}^T$ and binary masks $\mathcal{M}_{1:T}$ with $\mathcal{M}_t \subseteq [W] \times [H]$ for $t = 1, \ldots, T$ over the observable grid. Executing $\mathbf{a}$ in the environment produces the observation trajectory $\tau = (o_0, o_1, \ldots, o_T)$.

Masked-frame-prediction asks the agent to predict the colors in the masked region of the final observation, $o_T|_{\mathcal{M}}$, choosing from $K$ candidates $\mathcal{C}_{\text{choices}} = \{c_1, c_2, \ldots, c_K\}$, where each $c_i \in \mathcal{C}^{|\mathcal{M}|}$ and $\mathcal{C}$ denotes the set of possible grid cell colors.

We construct each masked frame prediction task so that, for any realization of $\tau$, exactly one candidate $c^* \in \mathcal{C}_{\text{choices}}$ matches the ground truth, $c^* = o_T|_{\mathcal{M}}$. This ensures a unique correct answer even when the underlying POMDP dynamics are stochastic.

We use a simple binary accuracy metric: score $= \mathbb{1}[\text{correct}]$.

## C.2. Change Detection

We define each change detection task with an original environment $\mathcal{M}$ and a changed environment $\mathcal{M}'$, both written in the Autumn language. We write their transition and observation functions as $(\mathcal{T}, \Omega)$ and $(\mathcal{T}', \Omega')$. A boolean guard $g(s, a)$ in an Autumn event handler gates the change in $\mathcal{M}'$; $g(s, a)$ determines when $\mathcal{T}'$ deviates from $\mathcal{T}$.

We fix an initial state distribution and an action sequence $\mathbf{a} = (a_1, \ldots, a_T)$. Executing $\mathbf{a}$ in $\mathcal{M}'$ produces a realized observation prefix $o'_{0:T}$. We define the change time $t^*$ as the first step where this realized prefix leaves the support of $\mathcal{M}$:

$$t^* = \min\left\{ t \geq 1 : \underbrace{\mathbb{P}_{\mathcal{M}}(o_{0:t} = o'_{0:t} \mid a_{1:t})}_{=0} \text{ and } \underbrace{\mathbb{P}_{\mathcal{M}}(o_{0:t-1} = o'_{0:t-1} \mid a_{1:t-1})}_{>0} \right\}.$$

Equivalently, $t^*$ is the earliest time when $o'_{0:t}$ has zero probability under $\mathcal{M}$ while $o'_{0:t-1}$ still has positive probability under $\mathcal{M}$. We require the guard to trigger exactly at $t^*$: $g(s'_{t^*-1}, a_{t^*}) = \text{true}$ and $g(s'_{t-1}, a_t) = \text{false}$ for all $t < t^*$.

The agent outputs a time index $t$ marking the detected change.

$$\text{score}(t) = \begin{cases} 0 & \text{if } t < (t^* - 1), \\ 1 & \text{if } t \in \{t^* - 1, \ t^*\}, \\ 1.377 \cdot f_{t^*}(t) - 1.178 & \text{otherwise.} \end{cases}$$

We set $f_{t^*}(t) = \frac{1}{1 - \frac{t}{t^*} \cdot e^{-\frac{t}{t^*}}}$; $t^*$ denotes the change time. This rule gives full credit for exact or one-step-early detection and smoothly penalizes late detections.

## C.3. Planning

We define each planning task by a goal specification $(\mathcal{S}, g)$, where $\mathcal{S} \subseteq [W] \times [H]$ indexes a subgrid, and $g \in \mathcal{C}^{|\mathcal{S}|}$ specifies the target colors on $\mathcal{S}$. The agent must produce an action sequence $\mathbf{a}$ that, when executed, yields a final observation $o_T$ matching the goal on the subgrid, that is, $o_T|_{\mathcal{S}} = g$. We use a simple binary success metric: score $= \mathbb{1}[o_T|_{\mathcal{S}} = g]$.

# D. Implementation Details

This section provides detailed definitions of the behavioral metrics and interface implementations used in our experiments. We first define normalized perplexity and the associated area under the curve (AUC), which quantify how agents' actions become more focused over time. We then describe the task-specific interface implementations for both the graphical user interface that human participants use and the text-based interface that reasoning models use.

### D.1. Normalized Perplexity

**Definition D.1 (Normalized Perplexity)** *Let $p(a)$ be the empirical action distribution over an active alphabet of size $K$ within a sliding window. Given entropy $H(p) = -\sum_a p(a) \log_2 p(a)$ and perplexity $P = 2^{H(p)}$, normalized perplexity is:*

$$Perplexity_{norm} = \frac{P - 1}{K - 1} \in [0, 1]$$

*where $K$ is the number of unique actions observed in the window, including directional keys and click positions.*

**Definition D.2 (Area Under the Curve (AUC) of Normalized Perplexity)** *The AUC of normalized perplexity is computed as:*

$$AUC = \int_0^1 Perplexity_{norm}(x)\, dx$$

*where $x \in [0, 1]$ is the normalized interaction position.*

**Definition D.3 (Final Normalized Perplexity)** *The final normalized perplexity is the value of $Perplexity_{norm}$ evaluated at the last window of the interaction sequence.*

### D.2. Task Interface Implementation

We implemented distinct interfaces for human participants and reasoning models. Human participants interact with environments through a graphical user interface (GUI), while reasoning models use a text-based interface. The following sections detail each interface type across the three task families.

#### D.2.1. GRAPHICAL USER INTERFACE

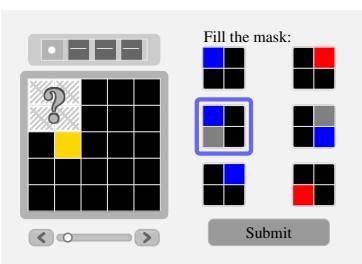

*Figure D.1.* Example of masked frame prediction task on the AutumnBench GUI showing the slider, action viewer, six choice options, and a `Submit` button.

For the masked frame prediction task, we implemented a slider to visualize the partially masked sequence of observations, as shown in Figure D.1. The GUI displays six answer choices from which participants select their prediction. Participants submit their selection using a `Submit` button.

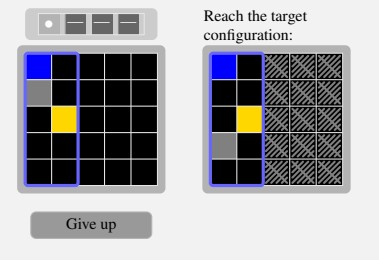

*Figure D.7.* Example of planning task on the AutumnBench GUI that shows a target region by masking out cells that are not in the target.

For the planning task, we display the goal state on a static grid alongside the interactive grid, with non-target regions shaded out, as shown in Figure D.7.

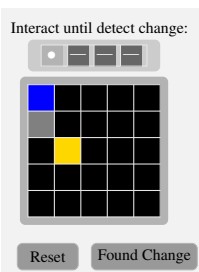 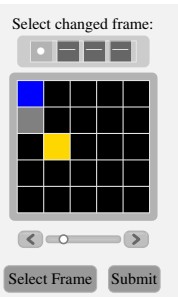

*Figure D.8.* Change detection on the AutumnBench GUI. (a) Initial change detection phase with `Reset` and `Found change` buttons. (b) Frame selection phase with the action viewer, slider, and `Submit` button.

For the change detection task, we provide a `Found change` button below the interactive grid, as shown in Figure D.8. Clicking this button transitions participants to the selection phase, where they use a slider to review observations from the test phase and identify the earliest changed frame. Participants submit their selection using a `Submit` button.

### D.2.2. TEXT-BASED INTERFACE

All reasoning models receive the same text-based prompts, with the system prompt shown in Listing D.1.

For the masked frame prediction task, we add two actions in the test phase: `step` and `rewind`, which allow the model to navigate through the partially masked observations. We provide six answer options as matrices of color strings, which the model selects using the `choose_option_i` action.

For the change detection task, we provide a `Found the change` action during the test phase. Once the model selects this action, we present `choose_grid_i` actions that enumerate the observations from the test phase. After the model chooses a grid, we display it in the subsequent prompt, allowing the model to confirm its selection using the `submit` action.

For the planning task, we provide the target configuration in the same format as the grid observation at every timestep of the test phase.

Listings D.2 to D.4 give example prompts for all three task types.

```
You are a helpful assistant currently operating as a curious agent exploring an environment that consists of a grid
    containing cells which can take colors.
You will be given observations and available actions to choose from at each step.
Your task is to interact with the environment efficiently and effectively and try to understand the underlying rules
    of the environment.

Here is a description of the actions:

- 'click x y' - Click on the cell at the location (x, y) on the grid.
- 'left' - Press the left arrow key.
- 'right' - Press the right arrow key.
- 'up' - Press the up arrow key.
- 'down' - Press the down arrow key.
- 'noop' - Do nothing and continue to the next step.
- 'quit' - Quit the environment.
- 'step' - Step through a sequence one frame at a time.
- 'go-to-test' - Go to the test phase.
- 'reset' - Reset the environment to the initial state.

Additional actions will be described whenever available.

Follow exactly the format when producing the action. So if the action is to click on a cell at location (1, 2), you
    should provide the action as <action>click 1 2</action>.
```

*Listing D.1.* System prompt for all reasoning models

```
[
    {
      "role": "user",
      "content": "Welcome, you are now in the interactive phase, where you can interact with the grid using the
      available actions.\nDuring the interactive phase your goal is to act in the environment to understand the
      underlying rules of the environment. \nUnderstand the environment and the dynamics of the environment well.
      Once you have understood the environment, you can select 'go-to-test' to go to the test phase.\nAfter the
      interactive phase you will be asked to use this knowledge about the environment to answer some questions about
      it.\n\n# Task Description:\nIn the test phase, you will step through frames from a trajectory in this same
      environment you interacted with (use the 'step' action to step through the trajectory). Each frame is
      structured as a json object with the following fields:\n\"render\": the grid observed,\n\"video_location\":
      timestep at which the frame was observed,\n\"action_took\": action taken at this timestep,\n\"is_finished\":
      whether the episode is finished.\nYou will step through the trajectory one frame at a time. Towards the end of
      the trajectory, parts of the grid will be masked (where the masked locations are marked as 'mask') and you will
      be given a set of choices to fill in the masked region at the final timestep. You need to choose option that
      fits the masked region at the final timestep.\n\nHere is the initial state of the grid: \n[[\"black\", \"
      black\", \"red\", ...], ...]\n\nThe following actions are available at this step: left,\nright,\nup,\ndown,\
      nclick [0-9] [0-9],\nnoop,\nquit,\ngo-to-test,\nreset\nThink step by step about the next action that should be
      taken. Remember, you are exploring the environment and trying to understand the underlying rules. Reflect on
      your action and self evaluate any potential issues before selecting the action. Output your final choice of
      action within a <action> tag.\nAdditionally, you can modify the contents of the scratchpad to use as memory
      since you can only observe the most recent states.\nPlease include the additions to the scratchpad withing <
      scratchpad_add> tags and deletions withing <scratchpad_del> tags. Output your choice of action within a <action
      > tag."
    },
    ...,
    {
      "role": "user",
      "content": "The interaction phase is now over. You will now step through frames from a trajectory in this
      same environment you interacted with (use the 'step' action to step through the trajectory). Each frame is
      structured as a json object with the following fields:\n\"render\": the grid observed,\n\"video_location\":
      timestep at which the frame was observed,\n\"action_took\": action taken at this timestep,\n\"is_finished\":
      whether the episode is finished.\nYou will step through the trajectory one frame at a time. Towards the end of
      the trajectory, parts of the grid will be masked (where the masked locations are marked as 'mask') and you will
      be given a set of choices to fill in the masked region at the final timestep. You need to choose option that
      fits the masked region at the final timestep. You can also use the 'rewind' action to go back to the previous
      frame.\n{\"video_location\": \"0/21\", \"render\": [[\"black\", \"black\", \"red\", \"black\", ...], ...], \"
      action_took\": \"start\", \"is_finished\": false}\n\nThe following actions are available at this step: step\
      nThink step by step about the next action that should be taken. Remember, you are exploring the environment and
      trying to understand the underlying rules. Reflect on your action and self evaluate any potential issues
      before selecting the action. Output your final choice of action within a <action> tag.\nAdditionally, you can
      modify the contents of the scratchpad to use as memory since you can only observe the most recent states.\
      nPlease include the additions to the scratchpad withing <scratchpad_add> tags and deletions withing <
      scratchpad_del> tags. Output your choice of action within a <action> tag."
    }
]
```

*Listing D.2.* Prompt for masked frame prediction task

```
[
    {
        "role": "user",
        "content": "Welcome, you are now in the interactive phase, where you can interact with the grid using
the available actions.\nDuring the interactive phase your goal is to act in the environment to understand the
underlying rules of the environment. You can reset the environment to it's initial state at any time.\
nUnderstand the environment and the dynamics of the environment well. Once you have understood the environment,
 you can select 'go-to-test' to go to the test phase.\nAfter the interactive phase you will be asked to use
this knowledge about the environment to answer some questions about it.\n\n# Task Description:\nIn the test
phase, you will interact with a changed version of the environment – where one of the dynamics rules has been
changed. \nYour goal is to use you understanding of the environemnt from the interaction phase to detect the
change. The environment will start in a normal state and then at some point, the environment will transition to
 a changed state. \nAs soon as you detect the change, you have to select 'I found the change!' action to go to
the next phase, wherein you have to choose exactly which frame the change occurred, then submit it. You may
choose as many times as you want to see the frames. You will be penalized if you click 'I found the change!'
before the change is detected. \n\n\nHere is the initial state of the grid: \n[[\"black\", \"black\", \"black
\", ...], ...]\n\nThe following actions are available at this step: left,\nright,\nup,\ndown,\nclick [0-15]
[0-15],\nnoop,\nquit,\ngo-to-test,\nreset\nThink step by step about the next action that should be taken.
Remember, you are exploring the environment and trying to understand the underlying rules. Reflect on your
action and self evaluate any potential issues before selecting the action. Output your final choice of action
within a <action> tag.\nAdditionally, you can modify the contents of the scratchpad to use as memory since you
can only observe the most recent states.\nPlease include the additions to the scratchpad withing <
scratchpad_add> tags and deletions withing <scratchpad_del> tags. Output your choice of action within a <action
> tag."
    },
    ...,
    {
        "role": "user",
        "content": "You are now in the test phase.You will now interact with a changed version of the
environment – where one of the dynamics rules has been changed. Your goal is to use you understanding of the
environemnt from the interaction phase to detect the change. The environment will start in a normal state and
then at some point, the environment will transition to a changed state. As soon as you detect the change, you
have to select 'I found the change!' action to go to the next phase, wherein you have to choose exactly which
frame the change occurred, then submit it. You may choose as many times as you want to see the frames. You will
 be penalized if you click 'I found the change!' before the change is detected. Here is the initial frame: [[\"
black\", \"black\", \"black\", ...], ...]\n\nThe following actions are available at this step: left,\nright,\
nup,\ndown,\nclick [0-15] [0-15],\nnoop,\nquit,\nI found the change!\nreset,\nquit\nThink step by step about
the next action that should be taken. Remember, you are exploring the environment and trying to understand the
underlying rules. Reflect on your action and self evaluate any potential issues before selecting the action.
Output your final choice of action within a <action> tag.\nAdditionally, you can modify the contents of the
scratchpad to use as memory since you can only observe the most recent states.\nPlease include the additions to
 the scratchpad withing <scratchpad_add> tags and deletions withing <scratchpad_del> tags. Output your choice
of action within a <action> tag."
    },
    ...
    {
        "role": "user",
        "content": "You are now in the test phase.You will now interact with a changed version of the
environment – where one of the dynamics rules has been changed. Your goal is to use you understanding of the
environemnt from the interaction phase to detect the change. The environment will start in a normal state and
then at some point, the environment will transition to a changed state. As soon as you detect the change, you
have to select 'I found the change!' action to go to the next phase, wherein you have to choose exactly which
frame the change occurred, then submit it. You may choose as many times as you want to see the frames. You will
 be penalized if you click 'I found the change!' before the change is detected. Here is the initial frame: [[\"
black\", \"black\", \"black\", ...], ...]\n\nThe following actions are available at this step: choose_frame_0,\
nchoose_frame_1,\nchoose_frame_2,\nSubmit choice,\nquit\nThink step by step about the next action that should
be taken. Remember, you are exploring the environment and trying to understand the underlying rules. Reflect on
 your action and self evaluate any potential issues before selecting the action. Output your final choice of
action within a <action> tag.\nAdditionally, you can modify the contents of the scratchpad to use as memory
since you can only observe the most recent states.\nPlease include the additions to the scratchpad withing <
scratchpad_add> tags and deletions withing <scratchpad_del> tags. Output your choice of action within a <action
> tag."
    }
]
```

*Listing D.3.* Prompts for change detection task

```
[
    {
        "role": "user",
        "content": "Welcome, you are now in the interactive phase, where you can interact with the grid using
the available actions.\nDuring the interactive phase your goal is to act in the environment to understand the
underlying rules of the environment. \nUnderstand the environment and the dynamics of the environment well.
Once you have understood the environment, you can select 'go-to-test' to go to the test phase.\nAfter the
interactive phase you will be asked to use this knowledge about the environment to answer some questions about
it.\n\n# Task Description:\nIn the test phase, you will be given a goal state and a highlight mask of the same
size as the grid where 1 indicates the region to be reached and 0 indicates the region to be ignored. \nYour
aim is to solve a planning task in the environment you interacted by reaching the goal state in the highlighted
 region.\nNote that you can no longer reset the environment, so plan carefully. You will be given the same
environment as you interacted with in the interaction phase, you need to interact with it to reach the goal
state in the highlighted region.\nYour grid will be checked against the goal state and the highlight mask at
every timestep. If you reach the goal state in the highlighted region, you will be given a reward. You may
choose to quit at any time if you are stuck.\n\n\nHere is the initial state of the grid: \n[[\"black\", \"black
\", \"black\", ...], ...]\n\nThe following actions are available at this step: left,\nright,\nup,\ndown,\nclick
 [0-15] [0-15],\nnoop,\nquit,\ngo-to-test,\nreset\nThink step by step about the next action that should be
taken. Remember, you are exploring the environment and trying to understand the underlying rules. Reflect on
your action and self evaluate any potential issues before selecting the action. Output your final choice of
action within a <action> tag.\nAdditionally, you can modify the contents of the scratchpad to use as memory
since you can only observe the most recent states.\nPlease include the additions to the scratchpad withing <
scratchpad_add> tags and deletions withing <scratchpad_del> tags. Output your choice of action within a <action
> tag."
    },
    ...,
    {
        "role": "user",
        "content": "The interaction phase is over, you have entered the test phase. You will now be given a goal
 state and a highlight mask of the same size as the grid where 1 indicates the region to be reached and 0
indicates the region to be ignored. \n               Your aim is to solve a planning task in the environment you
interacted by reaching the goal state in the highlighted region.\n             Note that you can no longer reset
 the environment, so plan carefully. You will be given the same environment as you interacted with in the
interaction phase, you need to interact with it to reach the goal state in the highlighted region.\n
 Your grid will be checked against the goal state and the highlight mask at every timestep. If you reach the
goal state in the highlighted region, you will be given a reward. You may choose to quit at any time if you are
 stuck.\n             The initial grid is:\n             {\"render\": [[\"black\", \"black\", \"black\", \"black
\", ...] ...], \"highlight_mask\": [[0, 1, 1, ...], ...]}\n\nThe following actions are available at this step:
left,\nright,\nup,\ndown,\nclick [0-15] [0-15],\nnoop,\nquit\nThink step by step about the next action
that should be taken. Remember, you are exploring the environment and trying to understand the underlying rules.
 Reflect on your action and self evaluate any potential issues before selecting the action. Output your final
choice of action within a <action> tag.\nAdditionally, you can modify the contents of the scratchpad to use as
memory since you can only observe the most recent states.\nPlease include the additions to the scratchpad
withing <scratchpad_add> tags and deletions withing <scratchpad_del> tags. Output your choice of action within
a <action> tag."
    }
]
```

*Listing D.4.* Prompts for planning task

# E. Theoretical Framework and Reference Baselines

Beyond the reasoning models evaluated in the main paper, interpreting AutumnBench scores requires reference points at both extremes: an oracle with perfect environment knowledge and an agent with none. Appendix E.1 first formalizes the three task types as next-state prediction problems, making explicit what a learned world model must do to solve each one; Section E.2.1 instantiates this framework with an Autumn-simulator agent that has ground-truth simulator access, setting an upper bound on what next-state prediction approaches can achieve. Section E.2.2 then characterizes a memoryless random agent. Together, these reference points frame how to read the human and model results in the main paper.

## E.1. Theoretical Framework for Next-State Prediction Models

This subsection formalizes the three AutumnBench task types in terms of a learned next-state prediction model. This framework clarifies how world-modeling approaches with next-frame prediction models would tackle each task and lays the groundwork for our Autumn-simulator agent baseline (Section E.2.1).

### E.1.1. NEXT-STATE PREDICTION MODEL

Let a next-state prediction model define $\hat{p}(x_{t+1} \mid x_{1:t}, a_{1:t})$.

**Masked Frame Prediction (MFP).** Select

$$\hat{c} \in \arg \max_{c \in \{1,\ldots,C\}} p\Big(x_{T+1} = x^{(c)} \mid x_1, a_{1:T}, x_{\bar{m},2:T+1}\Big),$$

with

$$p\Big(x_{T+1} = x^{(c)} \mid x_1, a_{1:T}, x_{\bar{m},2:T+1}\Big) := \int \mathbb{I}\Big[x_{T+1} = x^{(c)}\Big] \, p(x_{2:T+1} \mid x_1, a_{1:T}, x_{\bar{m},2:T+1}) \, dx_{2:T+1},$$

and

$$p(x_{2:T+1} \mid x_1, a_{1:T}, x_{\bar{m},2:T+1}) \propto \mathbb{I}[x_{2:T+1} \text{ matches } x_{\bar{m},2:T+1}] \, \hat{p}(x_{2:T+1} \mid x_1, a_{1:T}).$$

A Monte-Carlo estimator (dropping the normalizer $Z$) is

$$\hat{c} \approx \arg \max_c \sum_{i=1}^N \mathbb{I}\Big(x^{(c)}, x_{T+1}^{(i)}\Big) \cdot \mathbb{I}\Big(x_{2:T+1}^{(i)} \text{ matches } x_{\bar{m},2:T+1}\Big), \quad x_{2:T+1}^{(i)} \sim \hat{p}(\cdot \mid x_1, a_{1:T}).$$

I.e., we sample $N$ rollouts from $\hat{p}$, keep those consistent with the masks, and pick the candidate that appears most often at $T+1$.

**Change Detection (CD).** Detect the first changed frame when model surprise exceeds an entropy-scaled threshold:

$$t^\star := \min \Big\{ t : \ -\log \hat{p}(x_t \mid x_{1:t-1}, a_{1:t-1}) > \kappa \, \mathbb{H}[\hat{p}(\cdot \mid x_{1:t-1}, a_{1:t-1})] \Big\}.$$

Here, we tune $\kappa > 0$ on held-out data and $\mathbb{H}[p] = \mathbb{E}_p[-\log p]$.

**Planning (PL).** Given $x_t$, action space $\mathcal{A}$, and a goal specified by a partial observation, use online planning with selective replanning (e.g., MCTS over a learned model):

• Maintain/reuse a search tree rooted at the current $x_t$.

• Replan when uncertainty is high (e.g., surprise, small value margins) or a budget trigger is met.

• Run $N$ UCT simulations with a goal-based reward; act by highest-visit action; advance the subtree after stepping.

## E.2. Non-LLM Baseline Agents

To establish performance bounds, we evaluate two baseline agents: a simulator agent with perfect environment access (Section E.2.1) and a random agent (Section E.2.2) that acts without any learning or memory. These baselines contextualize reasoning model performance by showing what perfect world knowledge achieves versus purely unguided exploration.

### E.2.1. AUTUMN-SIMULATOR AGENT

The Autumn-simulator agent has direct access to the ground-truth Autumn program simulator for each environment. For masked-frame prediction and change detection, this agent provides an upper bound on what any next-state prediction approach, including learned models such as Dreamer-v3 (Hafner et al., 2025), could achieve on AutumnBench. For planning, the agent uses BFS, which is not necessarily optimal; a learned policy could plausibly outperform it, so the simulator agent serves only as a reference point for this task type. Despite its privileged access, the agent must still contend with stochasticity and long planning horizons. The agent observes environments through the same visual interface and task APIs as the reasoning models, ensuring a fair comparison.

**Task-specific solvers.**

- **Masked-frame prediction**: Roll out the simulator along the given test actions to the final timestep; choose the option whose masked region best matches the ground-truth final grid.

- **Change detection**: Simulate the original and test-phase dynamics under the same random action sequence and report the earliest timestep at which the observed grids differ.

- **Planning**: Perform BFS over the action space using the ground-truth simulator until a state satisfying the highlighted goal region is reached; return the first (shortest) solution found.

To mitigate stochasticity, we repeat each procedure 10,000 times and aggregate as above; for planning, we cap search depth and breadth at 10.

### E.2.2. RANDOM-AGENT BASELINE

To characterize difficulty of each of the tasks, we implement a random-agent baseline. The random agent is a memoryless policy that selects uniformly among six action types (click, up, down, left, right, and no-op) with no learning, adaptation, or access to prior observations. When the agent selects click, it samples a grid position uniformly. Because this agent cannot learn from experience, its performance reflects the ease of each environment under unguided exploration. We estimate random-agent success using approximately 50 rollouts per environment with a 1000-step timeout. For masked-frame prediction, the agent chooses answers uniformly over six options, yielding a chance probability of 1/6.

Table E.1 reports per-environment random-agent results. For change detection, we report the probability of triggering the change within 1000 timesteps; for planning, the probability of reaching the goal within 1000 timesteps.

Across our 43 change detection tasks, the random agent triggers changes with mean probability 0.80 in an average of 295 steps. Twenty-four environments trigger changes with probability 1.0 within 1000 steps. Five require 900 or more steps and two never trigger changes within the timeout.

Across our 43 planning tasks, the random agent reaches the goal with mean probability 0.399 in an average of 679 steps. Nine environments are solved with probability 1.0, while 13 remain unsolved at the timeout.

*Table E.1.* Random-agent baseline by environment. Environments 1–26 are deterministic; 27–43 are stochastic. Columns report the probability of triggering the change within 1000 timesteps (CD) and reaching the goal within 1000 timesteps (PL).

| # | Env | CD | PL | # | Env | CD | PL | # | Env | CD | PL |
|---|------|------|------|----|-------|------|------|----|-------|------|------|
| 1 | 27VWC | 0.94 | 0.16 | 16 | 4CKC2 | 0.72 | 0.55 | 31 | 76Z75 | 1.00 | 0.04 |
| 2 | KFQYT | 1.00 | 1.00 | 17 | 7WWW9 | 0.66 | 0.53 | 32 | NF5VZ | 1.00 | 1.00 |
| 3 | NRDF6 | 1.00 | 0.71 | 18 | N2NTD | 0.85 | 0.00 | 33 | 236VK | 1.00 | 0.20 |
| 4 | 6JKKA | 0.98 | 0.20 | 19 | YS322 | 0.26 | 0.00 | 34 | 6JVMF | 1.00 | 1.00 |
| 5 | K8MTQ | 1.00 | 0.00 | 20 | EAHCW | 1.00 | 0.00 | 35 | BT2KZ | 1.00 | 0.00 |
| 6 | T5F9B | 0.64 | 0.75 | 21 | QDHS3 | 0.36 | 0.00 | 36 | BY2Q7 | 0.00 | 0.00 |
| 7 | DQ8GC | 1.00 | 0.00 | 22 | VA6FQ | 0.77 | 0.06 | 37 | 4T8TR | 1.00 | 0.37 |
| 8 | QM9XB | 0.43 | 0.10 | 23 | 7VKTD | 1.00 | 0.55 | 38 | WHGHP | 0.00 | 0.22 |
| 9 | 27JBD | 1.00 | 0.00 | 24 | JXQAW | 1.00 | 0.55 | 39 | 83WKQ | 1.00 | 1.00 |
| 10 | VQJH6 | 1.00 | 0.94 | 25 | NTQ4Y | 0.53 | 0.00 | 40 | 3J4Z7 | 0.38 | 0.96 |
| 11 | QQM74 | 1.00 | 1.00 | 26 | DGG2C | 1.00 | 1.00 | 41 | XHGKQ | 0.13 | 0.00 |
| 12 | 7XF97 | 0.77 | 0.20 | 27 | VZ2Q4 | 1.00 | 0.02 | 42 | F5W3N | 0.89 | 0.94 |
| 13 | BT3GB | 1.00 | 0.00 | 28 | S2KT7 | 1.00 | 1.00 | 43 | 4N7BB | 1.00 | 0.04 |
| 14 | B8AKZ | 1.00 | 1.00 | 29 | B58F3 | 0.87 | 1.00 | | | | |
| 15 | 9F8AJ | 0.17 | 0.12 | 30 | N59TE | 1.00 | 0.00 | | | | |

# F. Additional Results

This section reports supporting analyses that accompany the main results. Appendix F.1 gives per-environment scores and examines how performance scales with computational cost and prompt choice. Appendix F.2 reports behavioral measurements during the interaction phase: action diversity, perplexity–score correlations, on-clause coverage, reset structure, and context-length scaling. Appendix F.3 reports context token usage per model. Appendix F.4 compares AutumnBench rankings with external benchmarks. Appendix F.5 presents representative agent traces.

A key finding is that despite complete access to ground-truth environment transitions, the simulator agent significantly underperforms humans on all three task types: masked-frame prediction, change detection, and planning. This gap highlights the need for better inference under uncertainty and long-horizon reasoning.

## F.1. Detailed Performance Analysis

This subsection presents comprehensive per-environment performance results (Section F.1.1), examines how reasoning models scale with computational resources (Section F.1.3), and assesses robustness to prompt variations (Section F.1.2).

### F.1.1. PER-ENVIRONMENT SCORES

Table F.2 presents average scores for all agents across the three task types (change detection, masked frame prediction, and planning) for each of the 43 AutumnBench environments. These granular results reveal substantial heterogeneity in task difficulty and agent capabilities: some environments (e.g., 7WWW9, 76Z75) are solved reliably across multiple task types, while others (e.g., 6JKKA, K8MTQ) remain challenging for all agents.

**Saturation check with Claude Opus 4.6.** To test whether AutumnBench remains challenging for newer frontier models, we evaluated Claude Opus 4.6 under the same protocol, prompts, and single-trajectory setup as the other reasoning models. Table F.2 lists scores for all AutumnBench problems. Claude Opus 4.6 achieves the highest overall score among reasoning models at 39.9% (CD: 19.8%, MFP: 44.2%, Planning: 55.8%), surpassing the previous best of 27.5% by o3. It is also the first model to score above zero on two environments where no prior model succeeded on any task type. Despite this improvement, 12 environments still yield zero scores across all tasks and models, and the human baseline of 93.5% remains substantially ahead, indicating that AutumnBench is not saturated by current frontier models.

### F.1.2. PROMPT ROBUSTNESS ANALYSIS

*Table F.1.* Performance of reasoning models under different prompts on a subset of AutumnBench problems

| Model | new_system_prompt | new_system_prompt_refined | prompt_with_tutorial | original |
|---|---|---|---|---|
| Claude 4 Sonnet | $0.1605 \pm 0.3389$ | $0.0630 \pm 0.2142$ | $0.1418 \pm 0.3442$ | $\mathbf{0.2609} \pm 0.4490$ |
| Gemini 2.5 Pro | $0.1880 \pm 0.3853$ | $\mathbf{0.1917} \pm 0.3890$ | $0.1667 \pm 0.3807$ | $0.1667 \pm 0.3807$ |
| o3 | $0.1539 \pm 0.3537$ | $0.1764 \pm 0.3834$ | $\mathbf{0.2214} \pm 0.3982$ | $0.1769 \pm 0.3846$ |

To assess the sensitivity of our results to prompt design, we evaluated three alternative prompt formulations alongside the original prompt on a subset of AutumnBench problems. The prompts and complete interaction logs are available in the dataset release (Anonymous, 2025). We tested: (1) new_system_prompt, which provides more explicit task instructions; (2) new_system_prompt_refined, an enhanced version that we generated using Claude's Prompt Improver tool with example interaction traces[1]; and (3) prompt_with_tutorial, which includes worked examples from the human-facing tutorial.

Table F.1 shows that the optimal prompt varies by model: Gemini 2.5 Pro achieves highest performance with new_system_prompt_refined, Claude 4 Sonnet with the original prompt, and o3 with prompt_with_tutorial. Differences across prompts remain modest (typically $\Delta < 0.1$), suggesting that model capabilities rather than prompt engineering determine performance. The evaluation subset comprises all three task types from eight environments: S2KT7, 27VWC, KFQYT, B8AKZ, 4CKC2, 4T8TR, QDHS3, and NTQ4Y.

---

[1]Claude Prompt Improver documentation

### F.1.3. PERFORMANCE VERSUS COMPUTATIONAL COST

To investigate if increased computational resources translate to better performance, we computed the cost-per-problem for each reasoning model. We then partition the 43 AutumnBench environments into two categories based on whether average agent performance improves monotonically with known computational cost: *cost-increasing* environments where performance strictly increases with computational budget, and *non-increasing* environments where performance plateaus or decreases despite additional resources.

**Overall findings.** Performance improves with higher computational cost in 25 of 43 environments (58%), but remains flat or decreases in 18 environments (42%). Critically, *no* environment achieves perfect performance (score = 1.0) with the least expensive model, indicating that some minimum capability threshold is necessary. The failure of 42% of environments to benefit from additional compute suggests fundamental reasoning limitations that scaling resources alone cannot overcome.

**Task-specific patterns.** Masked frame prediction shows improvements in 16 environments (37.2%), including `S2KT7`, `T5F9B`, `N59TE`, `NF5VZ`, `236VK`, `6JVMF`, `27JBD`, `BT2KZ`, `4CKC2`, `N2NTD`, `YS322`, `83WKQ`, `VA6FQ`, `XHGKQ`, `4N7BB`, and `DGG2C`. The remaining 62.7% show no improvement regardless of additional resources.

Change detection exhibits the least benefit from additional computation, with improvements in only 14 environments (32.6%): `NRDF6`, `N59TE`, `NF5VZ`, `236VK`, `27JBD`, `QQM74`, `BT2KZ`, `4CKC2`, `N2NTD`, `EAHCW`, `3J4Z7`, `7VKTD`, `VZ2Q4`, and `JXQAW`.

Planning demonstrates improvements in 16 environments (37.2%): `S2KT7`, `KFQYT`, `B58F3`, `27JBD`, `VQJH6`, `QQM74`, `9F8AJ`, `7WWW9`, `YS322`, `EAHCW`, `83WKQ`, `3J4Z7`, `7VKTD`, `F5W3N`, `JXQAW`, and `4N7BB`.

**Environment characteristics.** Certain environments (`27JBD`, `QQM74`, `7VKTD`, `JXQAW`) consistently benefit from additional resources across multiple tasks, suggesting they require sophisticated reasoning that scales with compute. Conversely, environments like `27VWC`, `6JKKA`, `K8MTQ`, and `76Z75` show no improvement in any task, indicating either very low or very high baseline difficulty that additional compute cannot address. More broadly, SetB environments cluster at difficulty extremes: they are either very easy, solvable with few random actions (see Table E.1), or very hard, requiring long-horizon reasoning about the dynamics.

*Table F.2.* Average scores by environment across three task types and agents. Qwen refers to Qwen3-235B-A22B-Thinking-2507.

| Env | Change Detection | | | | | | | Masked Frame Prediction | | | | | | | Planning | | | | | | |
|---|---|---|---|---|---|---|---|---|---|---|---|---|---|---|---|---|---|---|---|---|---|
| | Claude 4 Sonnet | Gemini 2.5 Pro | Human | o3 | Gemini 2.5 Flash | Qwen | Claude Opus 4.6 | Claude 4 Sonnet | Gemini 2.5 Pro | Human | o3 | Gemini 2.5 Flash | Qwen | Claude Opus 4.6 | Claude 4 Sonnet | Gemini 2.5 Pro | Human | o3 | Gemini 2.5 Flash | Qwen | Claude Opus 4.6 |
| S2KT7 | 0.0 | 0.0 | 0.86 | 0.0 | 0.0 | 0.0 | 0.3 | 1.0 | 1.0 | 1.0 | 1.0 | 0.0 | 1.0 | 1.0 | 1.0 | 0.0 | 1.0 | 1.0 | 1.0 | 1.0 | 1.0 |
| 27VWC | 0.0 | 0.0 | 0.99 | 0.0 | 0.0 | 0.0 | 0.3 | 0.0 | 0.0 | 0.20 | 0.0 | 1.0 | 0.0 | 0.0 | 0.0 | 0.0 | 0.66 | 0.0 | 0.0 | 0.0 | 0.0 |
| KFQYT | 0.0 | 0.0 | 0.93 | 0.0 | 0.0 | 0.0 | 0.0 | 0.0 | 1.0 | 1.0 | 1.0 | 1.0 | 0.0 | 0.0 | 0.0 | 0.0 | 1.0 | 1.0 | 0.0 | 0.0 | 1.0 |
| B58F3 | 0.0 | 0.0 | 0.99 | 0.0 | 0.0 | 0.0 | 1.0 | 1.0 | 0.0 | 1.0 | 1.0 | 1.0 | 0.0 | 1.0 | 1.0 | 0.0 | 1.0 | 1.0 | 0.0 | 0.0 | 0.0 |
| NRDF6 | 1.0 | 0.0 | 0.99 | 0.62 | 0.36 | 0.0 | 0.8 | 1.0 | 0.0 | 1.0 | 0.0 | 0.0 | 0.0 | 0.0 | 0.0 | 1.0 | 1.0 | 1.0 | 1.0 | 0.0 | 1.0 |
| 6JKKA | 0.0 | 0.0 | 1.0 | 0.0 | 0.0 | 0.0 | 0.0 | 0.0 | 0.0 | 0.38 | 0.0 | 0.0 | 0.0 | 0.0 | 0.0 | 0.0 | 1.0 | 0.0 | 0.0 | 0.0 | 0.0 |
| K8MTQ | 0.0 | 0.0 | 1.0 | 0.0 | 0.0 | 0.0 | 0.0 | 0.0 | 0.0 | 0.52 | 0.0 | 0.0 | 0.0 | 0.0 | 0.0 | 0.0 | 1.0 | 0.0 | 0.0 | 0.0 | 1.0 |
| T5F9B | 0.0 | 0.0 | 0.15 | 0.0 | 0.0 | 0.0 | 0.0 | 0.0 | 0.0 | 0.73 | 0.0 | 0.0 | 0.0 | 1.0 | 0.0 | 0.0 | 1.0 | 1.0 | 0.0 | 0.0 | 0.0 |
| N59TE | 0.0 | 0.0 | 1.0 | 0.80 | 0.0 | 0.0 | 0.0 | 0.0 | 0.0 | 0.98 | 0.0 | 0.0 | 0.0 | 1.0 | 1.0 | 1.0 | 1.0 | 1.0 | 0.0 | 0.0 | 0.0 |
| 76Z75 | 0.0 | 0.0 | 0.81 | 0.0 | 0.0 | 0.0 | 0.0 | 1.0 | 0.0 | 1.0 | 1.0 | 1.0 | 1.0 | 1.0 | 1.0 | 0.0 | 1.0 | 0.0 | 1.0 | 1.0 | 1.0 |
| NF5VZ | 0.0 | 1.0 | 0.88 | 0.0 | 0.0 | 0.0 | 0.0 | 0.0 | 1.0 | 0.68 | 0.0 | 1.0 | 0.0 | 0.0 | 0.0 | 0.0 | 0.0 | 0.0 | 0.0 | 0.0 | 1.0 |
| 236VK | 1.0 | 0.0 | 1.0 | 0.46 | 0.0 | 0.0 | 0.8 | 0.0 | 0.0 | 0.0 | 0.0 | 0.0 | 1.0 | 1.0 | 0.0 | 0.0 | 1.0 | 1.0 | 0.0 | 0.0 | 1.0 |
| DQ8GC | 0.0 | 1.0 | 1.0 | 0.0 | 0.95 | 0.95 | 0.9 | 1.0 | 0.0 | 0.0 | 0.0 | 0.0 | 0.0 | 0.0 | 0.0 | 0.0 | 1.0 | 0.0 | 0.0 | 0.0 | 1.0 |
| QM9XB | 0.0 | 0.0 | 1.0 | 0.0 | 0.0 | 0.0 | 0.0 | 0.0 | 0.0 | 0.98 | 0.0 | 0.0 | 0.0 | 0.0 | 0.0 | 0.0 | 1.0 | 0.0 | 0.0 | 0.0 | 0.0 |
| 6JVMF | 0.0 | 0.0 | 0.98 | 0.0 | 0.0 | 0.0 | 0.0 | 0.0 | 0.0 | 1.0 | 0.0 | 0.0 | 0.0 | 0.0 | 1.0 | 0.0 | 1.0 | 0.0 | 0.0 | 0.0 | 0.0 |
| 27JBD | 0.0 | 1.0 | 1.0 | 0.23 | 0.0 | 0.96 | 0.0 | 1.0 | 1.0 | 1.0 | 1.0 | 0.0 | 0.0 | 1.0 | 0.0 | 0.0 | 0.33 | 1.0 | 0.0 | 1.0 | 1.0 |
| VQJH6 | 0.0 | 0.0 | 1.0 | 0.0 | 0.0 | 0.0 | 0.0 | 1.0 | 0.0 | 1.0 | 1.0 | 0.0 | 0.0 | 1.0 | 1.0 | 0.0 | 1.0 | 0.0 | 0.0 | 0.0 | 1.0 |
| QQM74 | 0.0 | 0.0 | 0.99 | 0.99 | 0.0 | 0.0 | 0.0 | 1.0 | 0.0 | 1.0 | 1.0 | 0.0 | 0.0 | 1.0 | 1.0 | 0.0 | 0.66 | 0.0 | 0.0 | 1.0 | 1.0 |
| 7XF97 | 0.0 | 0.0 | 0.64 | 0.0 | 0.0 | 0.0 | 0.0 | 0.0 | 0.0 | 0.60 | 0.0 | 0.0 | 0.0 | 0.0 | 0.0 | 0.0 | 0.66 | 0.0 | 0.0 | 0.0 | 0.0 |
| BT3GB | 0.0 | 0.0 | 1.0 | 0.0 | 0.98 | 0.0 | 0.0 | 0.0 | 0.0 | 0.98 | 0.0 | 0.0 | 0.0 | 0.0 | 0.0 | 0.0 | 1.0 | 0.0 | 0.0 | 0.0 | 0.0 |
| BT2KZ | 0.0 | 0.0 | 1.0 | 0.62 | 0.0 | 0.0 | 0.0 | 0.0 | 0.0 | 1.0 | 0.0 | 0.0 | 1.0 | 1.0 | 1.0 | 1.0 | 1.0 | 1.0 | 0.0 | 0.0 | 1.0 |
| B8AKZ | 0.0 | 0.0 | 1.0 | 0.99 | 1.0 | 0.0 | 0.0 | 0.0 | 0.0 | 0.90 | 0.0 | 0.0 | 0.0 | 0.0 | 0.0 | 0.0 | 1.0 | 0.0 | 0.0 | 0.0 | 0.0 |
| 9F8AJ | 0.0 | 0.0 | 1.0 | 0.0 | 0.0 | 0.0 | 0.0 | 1.0 | 0.0 | 0.55 | 0.0 | 0.0 | 0.0 | 0.0 | 0.0 | 0.0 | 1.0 | 0.0 | 0.0 | 0.0 | 1.0 |
| 4CKC2 | 1.0 | 0.0 | 1.0 | 0.0 | 0.0 | 0.0 | 0.0 | 1.0 | 1.0 | 0.57 | 0.0 | 0.0 | 0.0 | 0.0 | 1.0 | 1.0 | 1.0 | 0.0 | 1.0 | 1.0 | 1.0 |
| 7WWW9 | 0.0 | 0.0 | 1.0 | 0.0 | 1.0 | 0.0 | 0.0 | 1.0 | 1.0 | 1.0 | 1.0 | 1.0 | 0.0 | 0.0 | 1.0 | 1.0 | 1.0 | 1.0 | 1.0 | 1.0 | 1.0 |
| N2NTD | 0.0 | 1.0 | 0.99 | 0.0 | 0.0 | 0.0 | 1.0 | 0.0 | 0.0 | 0.60 | 0.0 | 0.0 | 0.0 | 0.0 | 0.0 | 0.0 | 1.0 | 1.0 | 0.0 | 0.0 | 1.0 |
| BY2Q7 | 0.0 | 0.0 | 0.74 | 0.0 | 0.0 | 0.0 | 0.0 | 0.0 | 0.0 | 0.69 | 0.0 | 0.0 | 0.0 | 0.0 | 0.0 | 0.0 | 1.0 | 0.0 | 0.0 | 0.0 | 0.0 |
| 4T8TR | 1.0 | 1.0 | 1.0 | 0.89 | 0.98 | 0.44 | 0.0 | 0.0 | 0.0 | 1.0 | 0.0 | 0.0 | 0.0 | 0.0 | 0.0 | 0.0 | 1.0 | 0.0 | 0.0 | 1.0 | 1.0 |
| YS322 | 0.0 | 0.0 | 1.0 | 0.0 | 0.0 | 0.0 | 1.0 | 0.0 | 1.0 | 0.94 | 0.0 | 0.0 | 0.0 | 1.0 | 1.0 | 0.0 | 1.0 | 0.0 | 0.0 | 0.0 | 1.0 |
| WHGHP | 0.0 | 0.0 | 0.97 | 0.0 | 0.0 | 0.0 | 0.0 | 0.0 | 0.0 | 0.91 | 0.0 | 0.0 | 0.0 | 0.0 | 0.0 | 0.0 | 1.0 | 0.0 | 0.0 | 0.0 | 0.0 |
| EAHCW | 0.0 | 0.0 | 1.0 | 0.33 | 0.0 | 0.0 | 0.9 | 0.0 | 0.0 | 0.84 | 0.0 | 0.0 | 0.0 | 1.0 | 0.0 | 0.0 | 0.66 | 0.0 | 0.0 | 0.0 | 1.0 |
| 83WKQ | 0.0 | 0.0 | 0.99 | 0.0 | 0.89 | 0.0 | 0.0 | 1.0 | 0.0 | 1.0 | 1.0 | 0.0 | 0.0 | 1.0 | 1.0 | 0.0 | 1.0 | 0.0 | 0.0 | 1.0 | 1.0 |
| 3J4Z7 | 0.0 | 0.0 | 0.93 | 0.96 | 0.0 | 0.0 | 0.0 | 0.0 | 0.0 | 1.0 | 1.0 | 0.0 | 0.0 | 1.0 | 0.0 | 0.0 | 1.0 | 0.0 | 0.0 | 0.0 | 1.0 |
| QDHS3 | 0.0 | 0.0 | 1.0 | 0.0 | 0.0 | 0.0 | 0.0 | 0.0 | 0.0 | 0.93 | 0.0 | 0.0 | 0.0 | 0.0 | 0.0 | 0.0 | 1.0 | 0.0 | 0.0 | 0.0 | 0.0 |
| VA6FQ | 0.0 | 0.0 | 1.0 | 0.0 | 0.0 | 0.0 | 0.0 | 0.0 | 0.0 | 0.33 | 0.0 | 0.0 | 0.0 | 0.0 | 0.0 | 0.0 | 1.0 | 0.0 | 0.0 | 0.0 | 0.0 |
| 7VKTD | 0.0 | 0.0 | 1.0 | 0.50 | 0.0 | 0.0 | 0.0 | 1.0 | 0.0 | 0.95 | 1.0 | 0.0 | 0.0 | 0.0 | 0.0 | 0.0 | 1.0 | 0.0 | 0.0 | 1.0 | 1.0 |
| VZ2Q4 | 0.0 | 0.0 | 1.0 | 0.35 | 0.0 | 0.0 | 0.0 | 1.0 | 1.0 | 1.0 | 0.0 | 1.0 | 0.0 | 0.0 | 1.0 | 0.0 | 1.0 | 0.0 | 0.0 | 0.0 | 0.0 |
| XHGKQ | 0.0 | 0.0 | 1.0 | 0.0 | 0.0 | 0.0 | 0.0 | 0.0 | 0.0 | 0.85 | 0.0 | 0.0 | 1.0 | 1.0 | 1.0 | 1.0 | 1.0 | 1.0 | 0.0 | 0.0 | 1.0 |
| F5W3N | 0.0 | 0.0 | 0.78 | 0.0 | 0.0 | 0.0 | 0.0 | 1.0 | 0.0 | 0.19 | 0.0 | 1.0 | 1.0 | 1.0 | 1.0 | 0.0 | 1.0 | 0.0 | 0.0 | 0.0 | 0.0 |
| JXQAW | 0.0 | 0.0 | 1.0 | 0.96 | 0.63 | 0.97 | 0.0 | 1.0 | 0.0 | 1.0 | 1.0 | 1.0 | 0.0 | 0.0 | 1.0 | 0.0 | 1.0 | 0.0 | 0.0 | 0.0 | 0.0 |
| 4N7BB | 0.0 | 0.0 | 0.94 | 0.76 | 0.95 | 0.0 | 0.7 | 1.0 | 1.0 | 1.0 | 1.0 | 1.0 | 1.0 | 1.0 | 1.0 | 0.0 | 1.0 | 0.0 | 0.0 | 0.0 | 0.0 |
| NTQ4Y | 0.0 | 0.0 | 0.31 | 0.0 | 0.0 | 0.0 | 0.8 | 0.0 | 0.0 | 0.98 | 0.0 | 0.0 | 0.0 | 1.0 | 0.0 | 0.0 | 1.0 | 0.0 | 0.0 | 0.0 | 1.0 |
| DGG2C | 0.0 | 0.0 | 0.99 | 0.0 | 0.99 | 0.0 | 0.0 | 0.0 | 0.0 | 1.0 | 0.0 | 0.0 | 1.0 | 1.0 | 1.0 | 0.0 | 1.0 | 0.0 | 0.0 | 0.0 | 0.0 |

## F.2. Behavioral Analysis

This section reports the behavioral measurements that accompany the aggregate scores in Section 5.2: per-environment action diversity, on-clause coverage, reset structure, and context-length scaling.

### F.2.1. ACTION-DISTRIBUTION ANALYSIS

*Table F.3.* Unique clicks and directional actions by environment and agent. Qwen refers to Qwen3-235B-A22B-Thinking-2507.

| Env | Average Number of unique clicks | | | | | | Average Number of unique directional actions | | | | | |
|---|---|---|---|---|---|---|---|---|---|---|---|---|
| | Claude 4 Sonnet | Gemini 2.5 Pro | o3 | Human | Gemini 2.5 Flash | Qwen | Claude 4 Sonnet | Gemini 2.5 Pro | o3 | Human | Gemini 2.5 Flash | Qwen |
| S2KT7 | 6.0 | 7.0 | 6.0 | 8.3 | 8.3 | 4.7 | 4.0 | 4.0 | 3.7 | 2.7 | 2.7 | 3.3 |
| 27VWC | 9.7 | 6.7 | 11.0 | 9.7 | 9.0 | 6.0 | 4.0 | 4.0 | 3.7 | 1.3 | 4.0 | 3.5 |
| KFQYT | 4.3 | 13.0 | 5.3 | 13.3 | 9.3 | 4.7 | 3.3 | 4.0 | 3.7 | 4.0 | 4.0 | 3.3 |
| B58F3 | 2.7 | 1.0 | 2.3 | 1.0 | 5.7 | 2.3 | 4.0 | 4.0 | 3.7 | 1.3 | 3.7 | 4.0 |
| NRDF6 | 14.3 | 13.3 | 8.3 | 19.3 | 8.3 | 7.7 | 4.0 | 3.3 | 1.7 | 3.7 | 3.7 | 3.0 |
| 6JKKA | 9.0 | 9.0 | 5.0 | 2.7 | 6.7 | 2.0 | 4.0 | 4.0 | 4.0 | 2.7 | 3.0 | 3.3 |
| K8MTQ | 10.0 | 6.3 | 7.3 | 26.3 | 6.7 | 8.0 | 4.0 | 3.7 | 3.0 | 3.3 | 4.0 | 4.0 |
| T5F9B | 8.7 | 10.0 | 11.3 | 19.0 | 4.7 | 6.7 | 3.7 | 2.7 | 2.3 | 2.0 | 4.0 | 3.0 |
| N59TE | 9.3 | 26.7 | 9.7 | 15.3 | 7.3 | 9.7 | 3.3 | 4.0 | 3.3 | 3.3 | 2.7 | 2.3 |
| 76Z75 | 4.7 | 5.7 | 4.7 | 17.7 | 5.0 | 3.3 | 4.0 | 4.0 | 4.0 | 4.0 | 4.0 | 3.3 |
| NF5VZ | 8.3 | 10.7 | 7.0 | 4.0 | 4.3 | 1.5 | 4.0 | 4.0 | 3.7 | 2.3 | 4.0 | 4.0 |
| 236VK | 4.7 | 4.3 | 2.7 | 6.7 | 8.0 | 3.3 | 4.0 | 4.0 | 4.0 | 3.7 | 2.3 | 4.0 |
| DQ8GC | 1.3 | 2.0 | 0.7 | 19.7 | 2.7 | 1.3 | 3.7 | 3.7 | 3.7 | 4.0 | 4.0 | 4.0 |
| QM9XB | 5.3 | 3.3 | 8.7 | 9.0 | 3.0 | 2.7 | 4.0 | 4.0 | 3.7 | 2.7 | 3.0 | 3.7 |
| 6JVMF | 4.0 | 5.7 | 2.0 | 3.7 | 1.0 | 1.5 | 4.0 | 4.0 | 4.0 | 3.7 | 4.0 | 4.0 |
| 27JBD | 5.7 | 7.0 | 7.3 | 18.7 | 10.7 | 3.3 | 3.0 | 1.7 | 1.7 | 2.7 | 1.7 | 2.7 |
| VQJH6 | 17.7 | 7.7 | 8.7 | 20.3 | 7.3 | 4.5 | 4.0 | 3.7 | 3.7 | 4.0 | 4.0 | 4.0 |
| QQM74 | 4.3 | 6.7 | 5.7 | 18.7 | 4.0 | 5.0 | 4.0 | 3.3 | 3.0 | 3.7 | 4.0 | 3.0 |
| 7XF97 | 18.7 | 18.3 | 9.0 | 22.7 | 7.7 | 7.7 | 4.0 | 4.0 | 4.0 | 4.0 | 4.0 | 2.3 |
| BT3GB | 3.7 | 1.0 | 4.0 | 6.0 | 4.3 | 0.5 | 4.0 | 4.0 | 4.0 | 3.7 | 4.0 | 3.5 |
| BT2KZ | 4.0 | 6.0 | 6.3 | 18.7 | 6.3 | 7.7 | 2.7 | 3.3 | 1.0 | 2.7 | 0.3 | 2.7 |
| B8AKZ | 3.7 | 8.7 | 4.0 | 22.0 | 3.7 | 6.0 | 4.0 | 4.0 | 4.0 | 4.0 | 4.0 | 4.0 |
| 9F8AJ | 9.7 | 20.3 | 4.3 | 7.7 | 10.3 | 6.7 | 2.7 | 4.0 | 3.0 | 2.7 | 4.0 | 2.0 |
| 4CKC2 | 9.0 | 5.7 | 7.0 | 13.0 | 5.0 | 7.0 | 2.7 | 2.7 | 2.3 | 2.7 | 2.7 | 2.3 |
| 7WWW9 | 3.0 | 5.0 | 3.7 | 10.3 | 2.7 | 2.0 | 4.0 | 4.0 | 4.0 | 3.0 | 4.0 | 4.0 |
| N2NTD | 8.3 | 6.3 | 4.3 | 12.0 | 6.3 | 9.0 | 4.0 | 4.0 | 4.0 | 2.7 | 4.0 | 3.7 |
| BY2Q7 | 3.3 | 11.0 | 11.3 | 18.7 | 9.3 | 9.0 | 2.7 | 4.0 | 2.7 | 4.0 | 3.7 | 3.0 |
| 4T8TR | 2.3 | 3.3 | 5.7 | 18.3 | 8.7 | 4.7 | 3.7 | 3.7 | 4.0 | 0.3 | 2.0 | 4.0 |
| YS322 | 13.7 | 23.3 | 9.7 | 75.7 | 12.7 | 7.0 | 3.3 | 4.0 | 2.3 | 4.0 | 3.3 | 1.5 |
| WHGHP | 2.3 | 6.7 | 2.0 | 7.0 | 3.3 | 3.3 | 4.0 | 4.0 | 4.0 | 4.0 | 3.7 | 4.0 |
| EAHCW | 13.3 | 9.3 | 15.7 | 55.0 | 5.3 | 8.7 | 3.3 | 4.0 | 4.0 | 4.0 | 2.7 | 2.7 |
| 83WKQ | 2.7 | 4.0 | 7.7 | 14.3 | 6.7 | 3.0 | 4.0 | 4.0 | 3.7 | 2.7 | 3.0 | 3.7 |
| 3J4Z7 | 16.0 | 4.3 | 9.7 | 6.0 | 4.3 | 3.5 | 4.0 | 4.0 | 3.0 | 1.3 | 2.0 | 4.0 |
| QDHS3 | 5.0 | 0.7 | 1.0 | 15.3 | 3.5 | 4.0 | 4.0 | 3.7 | 4.0 | 4.0 | 4.0 | 4.0 |
| VA6FQ | 14.3 | 3.0 | 13.7 | 33.3 | 3.7 | 7.3 | 4.0 | 4.0 | 3.7 | 4.0 | 4.0 | 2.7 |
| 7VKTD | 5.0 | 2.3 | 2.0 | 6.7 | 2.3 | 3.3 | 4.0 | 4.0 | 3.7 | 2.7 | 4.0 | 2.3 |
| VZ2Q4 | 3.3 | 3.3 | 9.0 | 6.0 | 2.7 | 5.0 | 4.0 | 3.0 | 2.3 | 1.3 | 2.7 | 2.7 |
| XHGKQ | 8.0 | 7.7 | 11.3 | 1.0 | 5.3 | 2.0 | 4.0 | 4.0 | 3.7 | 2.7 | 4.0 | 3.0 |
| F5W3N | 5.3 | 5.0 | 1.0 | 3.3 | 3.0 | 11.7 | 4.0 | 4.0 | 3.3 | 2.3 | 3.0 | 3.0 |
| JXQAW | 9.7 | 8.7 | 10.3 | 18.7 | 8.0 | 8.0 | 4.0 | 4.0 | 2.0 | 4.0 | 2.7 | 3.3 |
| 4N7BB | 8.0 | 6.0 | 6.3 | 9.0 | 5.7 | 5.5 | 4.0 | 4.0 | 2.3 | 2.7 | 3.0 | 1.0 |
| NTQ4Y | 9.3 | 8.0 | 10.3 | 42.0 | 7.0 | 6.3 | 3.7 | 2.7 | 2.0 | 2.7 | 3.3 | 2.3 |
| DGG2C | 2.3 | 3.7 | 9.7 | 4.0 | 2.0 | 4.0 | 4.0 | 4.0 | 2.3 | 3.7 | 3.7 | 3.0 |

Table F.3 reports the number of *unique* actions each agent takes during the interaction phase, separated into clicks and directional keys. Each distinct grid position counts as a separate click, so a 4-cell exploration contributes four unique clicks; repeated presses of the same directional key collapse to one, so pressing up ten times contributes one unique direction. This normalization controls for differences in grid size and interaction length.

Humans exercise more unique click positions than every reasoning model in every environment. The gap is widest in environments where the dynamics depend on which cell is clicked rather than on a small set of canonical actions. In YS322, humans produce 75.7 unique clicks on average while reasoning models produce between 7.0 and 23.3; in EAHCW, humans produce 55.0 while reasoning models produce between 5.3 and 15.7. Reasoning models press close to all four directional keys—up, down, left, right—across environments and keep doing so as interaction continues, while humans adapt their actions to those that result in changes in the environment.

### F.2.2. ON-CLAUSE COVERAGE

An agent that exercises more of an environment's causal structure should, in principle, build a better model of it. We quantify this with *on-clause coverage*: the fraction of an environment's conditional update rules—on clauses in its Autumn program—that are executed at least once during the interaction phase. An on clause is executed whenever its triggering condition holds, so coverage measures which dynamics the agent has actually observed rather than how diverse its action sequence is.

Humans attain the highest mean coverage at 92.5%, followed by Claude 4 Sonnet at 82.5%, o3 at 78.4%, Gemini 2.5 Pro at 74.2%, Gemini 2.5 Flash at 70.1%, and Qwen at 69.3%. Within each reasoning model, coverage correlates significantly

with test-phase score: Claude $r = 0.397$, $p = 0.008$; o3 $r = 0.387$, $p = 0.010$; Qwen $r = 0.344$, $p = 0.024$; Gemini 2.5 Flash $r = 0.307$, $p = 0.045$. Gemini 2.5 Pro trends in the same direction but does not reach significance at $r = 0.291$, $p = 0.058$. Human coverage is near-saturated, so within-human variation in score is uncorrelated with coverage at $r = 0.019$, $p = 0.902$.

### F.2.3. RESET SEQUENCE ANALYSIS

If an agent uses resets to test hypotheses, its pre- and post-reset action sequences should resemble each other: the agent replays a prefix with a small variation to isolate a rule. We measure this resemblance with the *longest common subsequence* (LCS) ratio between action sequences in fixed windows immediately before and after each reset. Values near 1 indicate near-identical replay; values near 0 indicate unstructured resets. For each interaction, we size the comparison window as $6\%$ of the interaction length, clipped to the range $[10, 50]$ actions, so the window scales with interaction length rather than being fixed in absolute steps. A fixed window would distort cross-agent comparisons because interaction lengths differ by an order of magnitude between humans and models. We include `no-op` actions, in which the agent deliberately waits a timestep, as regular entries in these sequences rather than filtering them out, since a `no-op` after a given prefix is itself an informative choice.

*Table F.4.* LCS ratio between pre- and post-reset action sequences with adaptive windows sized to $6\%$ of interaction length, clipped to $[10, 50]$ actions. Higher values indicate more structured resets. All human–model differences significant at $p < 10^{-18}$ under the Mann–Whitney U test.

| Agent | Mean | Median | Middle 95% |
|---|---|---|---|
| Human | 0.637 | 0.654 | [0.200, 1.000] |
| Gemini 2.5 Pro | 0.360 | 0.333 | [0.000, 1.000] |
| Claude 4 Sonnet | 0.347 | 0.317 | [0.100, 0.800] |
| Qwen | 0.325 | 0.300 | [0.000, 0.700] |
| Gemini 2.5 Flash | 0.273 | 0.200 | [0.000, 0.800] |
| o3 | 0.188 | 0.200 | [0.000, 0.500] |

Humans produce the most structured resets, with mean LCS 0.637 and median 0.654 in Table F.4. Reasoning-model means range from 0.188 for o3 to 0.360 for Gemini 2.5 Pro. Every pairwise human–model difference is significant at $p < 10^{-18}$ under the Mann–Whitney U test. Structured human resets are consistent with the deliberate hypothesis-testing behavior documented in cognitive science (Coenen et al., 2015; Bramley et al., 2018): humans reset in order to rerun a near-identical prefix and isolate the effect of a single variation.

### F.2.4. CONTEXT-LENGTH SCALING

A natural hypothesis is that models with more context budget should score higher, since interaction histories grow with exploration. We regress score on interaction length within each model, controlling for environment difficulty. Claude and Gemini 2.5 Pro occupy narrow token bands—$1\%$–$6\%$ and $1\%$–$16\%$ of their context windows respectively—leaving insufficient within-model variance for the test. The two models with wider interaction-length ranges yield a null effect for o3 at $\rho = 0.01$, $p = 0.94$, and a weakly positive effect for Qwen at $\rho = 0.20$, $p = 0.03$. Wang et al. (2026) predict that models trained on environment trajectories should benefit from longer context, but these zero-shot reasoning models show no such benefit here.

### F.3. Context Usage Analysis

Table F.5 reports the average and maximum input-token counts at the final test-phase turn for each model. Each API call includes the full interaction history (no truncation), so these counts reflect complete context.

*Table F.5.* Context usage per model. We measure tokens at the final test-phase turn, which includes the complete interaction history.

| Model | Window | Avg Tokens (% Window) | Max Tokens (% Window) |
|---|---|---|---|
| Claude 4 Sonnet | 1M | 13,271 (1.3%) | 62,601 (6.3%) |
| Gemini 2.5 Pro | 1M | 14,625 (1.5%) | 164,528 (16.5%) |
| Gemini 2.5 Flash | 1M | 64,752 (6.5%) | 228,067 (22.8%) |
| o3 | 200K | 46,629 (23.3%) | 147,444 (73.7%) |
| Qwen | 262K | 57,670 (22.0%) | 253,402 (96.7%) |

All models use far fewer tokens than their context windows permit. The API generates thinking tokens each turn but does not append them to the interaction history. For Claude and Gemini, thinking tokens stayed below 25% of the output limit; we did not capture traces for o3 and Qwen.

### F.4. Correlation with External Benchmarks

For construct validity, we ask whether per-model AutumnBench scores rank-correlate with scores on external benchmarks that plausibly draw on world modeling. Li et al. (2024) validate simulator benchmarks for robot manipulation by showing that in-simulator scores correlate with real-world policy performance; we apply the analogous check to AutumnBench. We correlate AutumnBench against SWE-bench Verified (Jimenez et al., 2024), TerminalBench (Merrill et al., 2026), GPQA Diamond (Rein et al., 2024), and AIME 2025 and provide the results in Table F.6. We took all external scores from public leaderboards as of 25 March 2026 and note agent scaffolds where applicable.

*Table F.6.* AutumnBench scores alongside external benchmark scores, with Spearman $\rho$ correlations. $^*p < 0.05$, $^{**}p < 0.001$.

| Model | AutumnBench | CD | MFP | PL | SWE-bench V. | GPQA Dia. | TerminalBench | AIME 2025 |
|---|---|---|---|---|---|---|---|---|
| Claude Opus 4.6 | **39.9%** | 19.8% | 44.2% | 55.8% | 75.6% | 89.6% | 81.8% | 94.2% |
| Claude 4 Sonnet | 25.6% | 9.5% | 30.2% | 34.9% | 74.6% | 83.4% | — | 74.3% |
| o3 | 27.5% | 22.2% | 27.9% | 32.6% | 58.4% | — | — | 88.3% |
| Gemini 2.5 Flash | 19.9% | 20.3% | 25.6% | 14.0% | 35.0% | 79.0% | 17.1% | 73.3% |
| Gemini 2.5 Pro | 19.4% | 11.6% | 25.6% | 20.9% | 53.6% | 84.4% | 32.6% | 87.7% |
| Qwen | 15.7% | 7.7% | 19.5% | 20.9% | 30.2% | 77.2% | — | 91.0% |
| Spearman $\rho$ (overall) | | | | | $0.89^*$ | 0.70 | 0.50 | 0.26 |
| Spearman $\rho$ (MFP) | | | | | $0.99^{**}$ | 0.82 | 0.87 | 0.20 |
| Spearman $\rho$ (PL) | | | | | $0.90^*$ | 0.67 | 1.00 | 0.55 |
| Spearman $\rho$ (CD) | | | | | 0.26 | 0.40 | −0.50 | −0.14 |

Overall AutumnBench scores correlate strongly with SWE-bench Verified ($\rho = 0.89$, $p < 0.05$), consistent with the hypothesis that world-modeling ability transfers to real-world software engineering tasks where models must learn environment dynamics from interaction. Among the three task types, MFP shows the tightest association with SWE-bench ($\rho = 0.99$, $p < 0.001$) and GPQA Diamond ($\rho = 0.82$), while Planning correlates with both SWE-bench ($\rho = 0.90$, $p < 0.05$) and TerminalBench ($\rho = 1.00$, $n = 3$). AIME 2025 shows negligible correlation with AutumnBench ($\rho = 0.26$), consistent with the expectation that mathematical competition performance relies on different capabilities than interactive world modeling. CD exhibits weak or negative correlations with all external benchmarks, which we attribute to CD's reliance on counterfactual reasoning—detecting how dynamics have *changed*—a skill that existing benchmarks do not directly assess. The small sample sizes, particularly for TerminalBench ($n = 3$), limit statistical power and warrant caution when interpreting individual coefficients.

### F.5. Qualitative Examples

To provide insight into agent reasoning processes, we present an example scratchpad trace from Claude 4 Sonnet on the `6JVMF` masked-frame prediction task. Reasoning models maintain scratchpads during both exploration and testing phases (see prompt specifications in Listings D.2 to D.4). These scratchpads reveal how models form and refine hypotheses about environment dynamics through interaction.

Listing F.1 shows Claude's complete scratchpad for an `6JVMF` episode. The trace illustrates several characteristic behaviors: (1) initial exploration with directional actions to understand basic mechanics; (2) boundary testing to identify edge cases; (3) hypothesis formation and revision as new observations arrive; (4) detailed frame-by-frame analysis during the test phase; and (5) final reasoning to select among the multiple-choice options provided in Listing F.2. The model demonstrates systematic exploration and explicit reasoning about environment state, though it sometimes exhibits repetitive behavior (e.g., boundary oscillation testing) that yields no additional information.

```
Initial observations:
- 16x16 grid, mostly black cells
- One blue cell at position (7, 7)
- Available actions: left, right, up, down, click [0-15] [0-15], noop, quit, go-to-test, reset
- Goal: Understand the underlying rules of the environment

Strategy:
1. Try different actions to see how they affect the grid
2. Start with arrow keys to see if they move something
3. Try clicking on different cells to see what happens
4. Look for patterns in how the environment responds
Observations after "right" action:
- Blue cell moved from (7, 7) to (6, 7) - moved LEFT when I pressed RIGHT
- This suggests arrow keys control the blue cell but in reverse direction
- Need to test other arrow keys to confirm this patternConfirmed pattern:
- LEFT key moves blue cell RIGHT
- RIGHT key moves blue cell LEFT
- Arrow keys control blue cell movement in reverse/opposite direction
- Blue cell moves horizontally along row 7 so far

Next tests needed:
- Test UP and DOWN keys to see vertical movement behavior
- Test boundaries - what happens when blue cell reaches edge?
- Test clicking on cells
Updated observations:
- UP key also moves blue cell LEFT (same as RIGHT key)
- Both UP and RIGHT move the blue cell left
- Need to test DOWN key to see if it follows the same pattern as LEFT
Complete movement pattern discovered:
- LEFT key moves blue cell RIGHT
- RIGHT key moves blue cell LEFT
- UP key moves blue cell LEFT
- DOWN key moves blue cell LEFT

So the controls are:
- Only LEFT moves the blue cell right
- RIGHT, UP, and DOWN all move the blue cell left
- Movement is only horizontal along row 7
- No vertical movement detected yet
Boundary behavior discovered:
- When blue cell reaches left edge (0, 7) and tries to move further left, it bounces back to (1, 7)
- Movement appears to be bounded by grid edges
- Need to test right boundary behavior as well
Additional boundary observations:
- When blue cell is at left edge (0, 7), pressing "left" moves it right to (1, 7)
- Movement controls work normally once away from the boundary
- Continuing to test right boundary behaviorClick behavior observations:
- Clicking on the blue cell at (0, 7) had no visible effect
- Clicking on adjacent cell (1, 7) had no visible effect
- Clicks don't seem to change the grid state or create new objects- Clicking on distant cell (8, 8) had no visible
      effect
- Clicks appear to have no effect on the environment at all
- Need to explore if vertical movement is possible, especially at boundaries
Boundary behavior at left edge:
- When at (0, 7), UP key moves blue cell right to (1, 7)
- This confirms UP still tries to move left but bounces at boundary
- Need to test DOWN key for potential vertical movement
DOWN key behavior confirmed:
- DOWN moves blue cell left (from (1,7) to (0,7))
- Still no vertical movement discovered
- Movement remains only horizontal along row 7
Boundary behavior observations:
- Blue cell seems to have inconsistent movement near left boundary
- Sometimes LEFT moves it right, sometimes it stays at (0,7)
- Need to test consistent movement patternsLeft boundary oscillation observed:
- Blue cell seems to oscillate between (0,7) and (1,7) when using LEFT key repeatedly
- May need different strategy to move away from left edge consistentlyBoundary oscillation pattern confirmed:
- At left boundary, blue cell oscillates between (0,7) and (1,7)
- All arrow keys (LEFT, RIGHT, UP, DOWN) seem to cause same oscillation behavior
- Need to try different strategy to break out of this patternClick behavior confirmed:
- Clicking on (8, 7) had no effect
- Clicks don't create new objects or change existing ones
- Blue cell remains trapped at left boundary

Next to try:
- "step" action to see if there's automatic progression
- "go-to-test" to see if there's a test phase
- Different exploration strategiesStep action tested:
- "step" action had no visible effect
- No automatic progression or animation detected
```

```
- Grid state remains unchanged

Summary of findings so far:
- Blue cell oscillates between (0,7) and (1,7) at left boundary
- All arrow keys cause same oscillation at boundary
- Clicks have no effect
- Step action has no effect
- Movement is only horizontal along row 7Test phase entered:
- "go-to-test" action changed available actions to only "noop"
- Grid state appears unchanged with blue cell at (0,7)
- This suggests a test/observation phase where interaction is limited
- May reveal automatic behavior or require demonstration of understandingTest phase trajectory observation:
Frame 0: Blue cell at (7, 7), action: "initial"
- This is the starting position, same as when I first entered the environment
- Need to observe how movement progresses through the trajectory
- Will track the blue cell's position and movement pattern
Frame 1: Blue cell at (7, 8), action: "noop"
- Blue cell moved DOWN from (7, 7) to (7, 8)
- Action was "noop" - no user input
- This reveals automatic downward movement/gravity!
- Different from exploration phase where I only saw horizontal movement
Frame 2: Blue cell at (7, 7), action: "noop"
- Blue cell moved UP from (7, 8) back to (7, 7)
- Action was "noop" again
- Pattern emerging: vertical oscillation between (7, 7) and (7, 8)
- Similar to boundary oscillation I saw during exploration
- Not consistent gravity, but oscillating movement with noop
Frame 3: Blue cell at (6, 7), action: "noop"
- Blue cell moved UP from (7, 7) to (6, 7)
- Continuing upward movement pattern
- Revised pattern: Initial down, then consistent upward movement with noop
- Need to see if this continues or if there are boundaries
Frame 4: Blue cell at (7, 7), action: "noop"
- Blue cell moved DOWN from (6, 7) back to (7, 7)
- Back to original position
- Pattern seems to be oscillating around (7, 7) position
- Movement sequence so far: center → down → center → up → center
- Need to continue observing to see if pattern repeats
Frame 5: Blue cell at (6, 7), action: "noop"
- Blue cell moved UP from (7, 7) to (6, 7)
- Same position as frame 3
- Pattern emerging: (7,7) → (7,8) → (7,7) → (6,7) → (7,7) → (6,7)
- Seems to be cycling through positions around the center
- Need to continue to confirm full cycle pattern
Frame 6: Blue cell at (7, 7), action: "noop"
- Blue cell moved DOWN from (6, 7) to (7, 7)
- Back at center position again
- Clear oscillation pattern emerging around (7, 7)
- Positions cycle: (7,7) → (7,8) → (7,7) → (6,7) → (7,7) → (6,7) → (7,7)
- Seems to alternate between moving to adjacent cells and returning to center
Frame 7: Blue cell at (6, 7), action: "noop"
- Blue cell moved UP from (7, 7) to (6, 7)
- Continuing the established oscillation pattern
- Pattern confirmed: alternates between (7,7) center and adjacent positions
- Movement sequence: down to (7,8), back to center, up to (6,7), back to center, repeat
Frame 8: Blue cell at (5, 7), action: "noop"
- Blue cell moved UP from (6, 7) to (5, 7)
- New position - breaking the previous oscillation pattern around (7,7)
- Movement is continuing upward instead of returning to center
- Pattern might be evolving or following a longer trajectory
- Need to continue observing to understand the full movement pattern
Frame 9: Blue cell at (4, 7), action: "noop"
- Blue cell moved UP from (5, 7) to (4, 7)
- Continuing consistent upward movement
- Pattern has shifted from oscillation to linear upward movement
- Movement is along column 7, going from row 7 toward row 0
- Need to see if it reaches the boundary or changes direction
Frame 10: Large masked region appeared
- Masked area covers rows 2-9, columns 4-11
- Blue cell was last seen at (4, 7) moving upward
- Based on movement pattern, blue cell should be at (3, 7) which is within masked region
- This is likely the test region I'll need to predict
- Need to continue stepping through remaining frames before making prediction
Frame 11: Action "click 7 4" performed
- Click action on position (7, 4) which is within masked region
- Grid appearance unchanged (still masked)
- This suggests trajectory includes click actions on masked positions
- Need to track all click actions to understand final prediction
- Position (7, 4) is where blue cell should be based on upward movement pattern
Frame 12: Action "noop"
- Grid appearance unchanged (still masked)
```

```
- Following the click action from previous frame
- Need to continue observing to see full sequence of actions
Frame 13: Action "noop"
- Grid appearance unchanged (still masked)
- Continuing sequence after the click action
- Need to continue to see if there are more actions or if we reach the final prediction frame
Frame 14: Action "noop"
- Grid appearance unchanged (still masked)
- Continuing sequence of noop actions after the click
- Need to continue to see if we reach final frame or more actions occur
Frame 15: Action "click 8 7" performed
- Second click action within masked region at position (8, 7)
- Grid appearance unchanged (still masked)
- Two click actions so far: (7, 4) and (8, 7)
- Need to continue to see if there are more actions or if we reach final prediction frame
Frame 16: Action "noop"
- Grid appearance unchanged (still masked)
- Following the second click action from previous frame
- Two click positions recorded: (7, 4) and (8, 7)
- Need to continue to see if we reach final frame
Frame 17: Action "noop"
- Grid appearance unchanged (still masked)
- Continuing sequence after both click actions
- Two click positions recorded: (7, 4) and (8, 7)
- Need to continue to see if we reach final prediction frame
Frame 18: Action "noop"
- Grid appearance unchanged (still masked)
- Continuing sequence of noop actions
- Two click positions remain: (7, 4) and (8, 7)
- Need to continue stepping to find final frame with options
Frame 19: Action "click 8 7" performed again
- Third click action within masked region, second time clicking (8, 7)
- Grid appearance unchanged (still masked)
- Click count: (7, 4) clicked once, (8, 7) clicked twice
- Need to continue to see if there are more actions or final frame
Frame 20: Action "noop"
- Grid appearance unchanged (still masked)
- Following the repeated click on (8, 7)
- Click count remains: (7, 4) clicked once, (8, 7) clicked twice
- Need to continue to see if we reach final frame with prediction options
Frame 21: Action "noop"
- Grid appearance unchanged (still masked)
- Continuing sequence of noop actions after second click on (8, 7)
- Click positions tracked: (7, 4) clicked once, (8, 7) clicked twice
- Need to continue stepping to reach final prediction frame
Frame 22: Blue cell appears at (8, 1)!
- First visible change since masked region appeared
- Blue cell is outside masked region at position (8, 1)
- Shares x-coordinate with clicked position (8, 7)
- This suggests click actions in masked region have effects outside it
- Need to continue to see if blue cell moves or if more changes occur
Frame 23: Second blue cell appears at (9, 1)!
- Now two blue cells: (7, 1) and (9, 1)
- Both at same row (y=1), different columns
- X-coordinates 7 and 9 relate to clicked positions (7, 4) and (8, 7)
- Pattern emerging: clicks in masked region create blue cells outside at row 1
- Possible transformation: x-coord preserved/shifted, y-coord becomes 1
Frame 24: Blue cell configuration changed dramatically
- Only one blue cell visible at (7, 0)
- Previous frame had two blue cells at (7, 1) and (9, 1)
- Blue cell at (7, 1) moved up to (7, 0)
- Blue cell at (9, 1) disappeared
- Pattern suggests upward movement and possible disappearance conditions
Frame 25: Blue cell moved to (8, 0)
- Blue cell moved from (7, 0) to (8, 0) - horizontal movement at top row
- Position (8, 0) aligns with x-coordinate of clicked position (8, 7)
- Movement pattern: vertical upward, then horizontal alignment
- Need to continue to see if blue cell continues moving or trajectory ends
Frame 26: New configuration with two blue cells
- Blue cell at (8, 1) - moved down from (8, 0) in previous frame
- Blue cell at (11, 5) - new appearance at edge of masked region
- Blue cells seem to be materializing and moving in relation to earlier click actions
- Need to continue to see if more blue cells appear or if masked region reveals more content
Frame 27: More blue cells appearing at masked region boundary
- Blue cell at (9, 1) - moved right from (8, 1)
- Blue cell at (11, 2) - new at right edge of masked region
- Blue cell at (11, 4) - another new at right edge of masked region
- Pattern: blue cells emerging at column 11 (right edge of mask) at various rows
- Movement continues for existing blue cells
Frame 28: Blue cell movements and new appearances
- Blue cell at (10, 1) - moved right from (9, 1)
```

```
- Blue cell at (12, 1) - new appearance at column 12
- Blue cell at (12, 3) - moved from (11, 2), shifted right and down
- Pattern: continued rightward movement and generation
- Some blue cells disappearing (the one at (11, 4) is gone)
Final analysis:
- Masked region is 8x8 (rows 2-9, cols 4-11)
- Click actions at (7,4) and (8,7) translate to (3,0) and (4,3) in masked region
- Option 0 has blue cells positioned consistently with click pattern
- Other options either too sparse, too dense, or misaligned with observed clicks
```

*Listing F.1.* Example scratchpad for the `6JVMF` environment, with choices listed in Listing F.2.

```
[
    [[\"blue\", \"black\", \"black\", \"black\", \"black\", \"black\", \"black\", \"black\"],
     [\"black\", \"black\", \"black\", \"blue\", \"black\", \"black\", \"black\", \"black\"],
     [\"black\", \"black\", \"black\", \"black\", \"black\", \"black\", \"blue\", \"black\"],
     [\"black\", \"black\", \"black\", \"black\", \"black\", \"black\", \"black\", \"black\"],
     [\"black\", \"black\", \"blue\", \"black\", \"black\", \"black\", \"black\", \"black\"],
     [\"black\", \"black\", \"black\", \"blue\", \"black\", \"black\", \"blue\", \"black\"],
     [\"black\", \"black\", \"blue\", \"black\", \"black\", \"black\", \"black\", \"black\"],
     [\"black\", \"black\", \"black\", \"blue\", \"black\", \"black\", \"black\", \"black\"]],

    [[\"black\", \"black\", \"black\", \"black\", \"black\", \"black\", \"black\", \"black\"],
     [\"black\", \"black\", \"black\", \"black\", \"black\", \"black\", \"black\", \"black\"],
     [\"black\", \"black\", \"black\", \"black\", \"black\", \"black\", \"black\", \"black\"],
     [\"black\", \"black\", \"blue\", \"black\", \"black\", \"black\", \"blue\", \"black\"],
     [\"black\", \"black\", \"black\", \"blue\", \"black\", \"black\", \"black\", \"black\"],
     [\"black\", \"black\", \"blue\", \"black\", \"black\", \"black\", \"black\", \"black\"],
     [\"black\", \"black\", \"black\", \"blue\", \"black\", \"black\", \"black\", \"black\"],
     [\"black\", \"black\", \"black\", \"black\", \"black\", \"black\", \"black\", \"black\"]],

    [[\"black\", \"black\", \"black\", \"black\", \"black\", \"blue\", \"black\", \"black\"],
     [\"blue\", \"black\", \"black\", \"blue\", \"black\", \"black\", \"black\", \"blue\"],
     [\"black\", \"black\", \"black\", \"black\", \"black\", \"black\", \"blue\", \"black\"],
     [\"black\", \"black\", \"black\", \"black\", \"black\", \"blue\", \"black\", \"black\"],
     [\"black\", \"black\", \"blue\", \"black\", \"black\", \"black\", \"black\", \"black\"],
     [\"black\", \"black\", \"black\", \"blue\", \"black\", \"black\", \"black\", \"black\"],
     [\"black\", \"black\", \"blue\", \"black\", \"black\", \"blue\", \"black\", \"black\"],
     [\"blue\", \"black\", \"black\", \"blue\", \"black\", \"black\", \"black\", \"black\"]],

    [[\"black\", \"black\", \"black\", \"black\", \"black\", \"black\", \"black\", \"black\"],
     [\"black\", \"black\", \"black\", \"black\", \"black\", \"black\", \"black\", \"black\"],
     [\"black\", \"black\", \"black\", \"black\", \"black\", \"black\", \"black\", \"black\"],
     [\"black\", \"black\", \"black\", \"black\", \"black\", \"black\", \"black\", \"black\"],
     [\"black\", \"black\", \"blue\", \"black\", \"black\", \"black\", \"black\", \"black\"],
     [\"black\", \"black\", \"black\", \"black\", \"black\", \"black\", \"black\", \"black\"],
     [\"black\", \"black\", \"black\", \"black\", \"black\", \"black\", \"black\", \"black\"],
     [\"black\", \"black\", \"black\", \"black\", \"black\", \"black\", \"black\", \"black\"]],

    [[\"black\", \"black\", \"black\", \"black\", \"black\", \"black\", \"black\", \"black\"],
     [\"black\", \"black\", \"black\", \"black\", \"black\", \"black\", \"black\", \"black\"],
     [\"black\", \"black\", \"black\", \"black\", \"black\", \"black\", \"black\", \"black\"],
     [\"black\", \"black\", \"black\", \"black\", \"black\", \"black\", \"black\", \"black\"],
     [\"black\", \"black\", \"blue\", \"blue\", \"black\", \"black\", \"black\", \"black\"],
     [\"black\", \"black\", \"blue\", \"black\", \"black\", \"black\", \"black\", \"black\"],
     [\"black\", \"black\", \"blue\", \"blue\", \"black\", \"black\", \"black\", \"black\"],
     [\"black\", \"black\", \"blue\", \"blue\", \"black\", \"black\", \"black\", \"black\"]],

    [[\"black\", \"black\", \"black\", \"black\", \"black\", \"black\", \"black\", \"black\"],
     [\"black\", \"black\", \"black\", \"black\", \"black\", \"black\", \"black\", \"black\"],
     [\"black\", \"black\", \"black\", \"black\", \"black\", \"black\", \"black\", \"black\"],
     [\"black\", \"black\", \"black\", \"black\", \"black\", \"black\", \"black\", \"black\"],
     [\"black\", \"black\", \"black\", \"black\", \"black\", \"black\", \"black\", \"black\"],
     [\"black\", \"black\", \"black\", \"black\", \"black\", \"black\", \"black\", \"black\"],
     [\"black\", \"black\", \"black\", \"black\", \"black\", \"black\", \"black\", \"black\"],
     [\"black\", \"black\", \"black\", \"black\", \"black\", \"black\", \"black\", \"black\"]]
]
```

*Listing F.2.* Answer choices, indexed 0–5, supplied for masked frame prediction in the 6JVMF environment.

