# OpenReview forum: "Benchmarking World-Model Learning with Environment-Level Queries"
_ICML.cc/2026/Conference — ICML 2026 regular_

### Official Review · Reviewer_UdVH · 2026-02-20

**Soundness:** 3
**Presentation:** 4
**Significance:** 4
**Originality:** 3
**Overall Recommendation:** 5
**Confidence:** 5

**Summary:**

This paper proposes a protocol to evaluate agents as implicit world models by pairing experiments with test queries.
43 environments, 129 tasks and 3 types of queries are instantiated in the test suite.
Experiments with humans and LLMs reveal a gap between humans and SOTA models in this setting.

**Compliance With Llm Reviewing Policy:**

Affirmed.

**Final Justification:**

After considering the paper and the authors’ rebuttal, I remain **supportive of acceptance**. I find the paper clear, well motivated, and significant: the proposed evaluation protocol for testing agents as implicit world models is interesting, timely, and supported by a broad experimental study across many environments and tasks.

My main questions concerned the connection to in-context learning / in-context RL, the role of context length, and a more detailed breakdown of model failure modes. The rebuttal addressed these points well by clarifying the conceptual relation to in-context RL, reporting concrete context-usage statistics, and adding useful analysis on exploration quality, state-space coverage, belief rigidity, and the limited benefit of longer context or higher-cost models. These responses strengthened the paper and reinforced my positive assessment.

Overall, I view this as a strong and important benchmark-style contribution with good clarity, solid experimental support, and clear relevance to the study of agent world models, and I maintain my accept recommendation.

**Key Questions For Authors:**

1. How much context does each experiment typically use?
2. Could you provide a more detailed analysis of the models’ failure modes—for example, whether performance is primarily limited by memory shrinkage, suboptimal exploration strategies, or failure to learn the underlying dynamics? In addition, do you expect that explicitly including the query types in the prompts would encourage better exploration?
3. How does the model’s success rate scale with context length, and do you observe any saturation or diminishing returns as the context grows?

**Limitations:**

Yes

**Strengths And Weaknesses:**

**Strengths:**
- The paper is clearly written and the problem (testing inner world models of agents interacting with the environments) is clear and important.
- The experiment setting meets the requirement of the problem, i.e. first explore, then exploit and reason about the world models.
- Scaled experiments demonstrate there still exist a gap between current LLMs and humans.

**Weaknesses:**
- There lacks literature discussions between the problem and in-context learning/in-context RL. Studies like _Context and Diversity Matter: The Emergence of In-Context Learning in World Models_ have shown that the learnt world model can get more accurate as context growths. I believe in-context RL suites this setting.

---

> ### Author Rebuttal · Authors · 2026-03-31
>
> ## R1. Adding a discussion of in-context learning and in-context RL to the related work, particularly in light of studies showing that learned world models improve with context length.
> Thank you for highlighting this. WorldTest's two-phase protocol (reward-free exploration then evaluation) shares structural features with in-context RL: both require extracting dynamics from interaction history, but WorldTest provides no reward signal, isolating world-model acquisition from policy optimization.
>
> One potential development direction for these agents is context engineering and in-context learning [1], where they must *learn* dynamics in-context rather than receiving them directly (as our simulator agent does). Such a model could generalize across environments from context alone, without retraining. Our simulator agent bounds what perfect next-frame knowledge can achieve, yet underperforms humans because stochasticity and long planning horizons remain challenging even with a perfect world model. We added this discussion to the related work.
> ## R2. Reporting how much context each experiment uses and providing a more detailed analysis of failure modes (memory shrinkage, suboptimal exploration, or failure to learn dynamics).
> ### Context usage
> All models use far fewer tokens than their windows permit. Each API call includes the full history (no truncation), so input tokens at the final test-phase turn reflect complete context.
>
> | Model | Window | Avg Tokens (% Window) | Max Tokens (% Window) |
> |---|---|---|---|
> | Claude 4 Sonnet | 1M | 13,271 (1.3%) | 62,601 (6.3%) |
> | Gemini 2.5 Pro | 1M | 14,625 (1.5%) | 164,528 (16.5%) |
> | Gemini 2.5 Flash | 1M | 64,752 (6.5%) | 228,067 (22.8%) |
> | o3 | 200K | 46,629 (23.3%) | 147,444 (73.7%) |
> | Qwen3-235B | 262K | 57,670 (22.0%) | 253,402 (96.7%) |
>
> Thinking tokens are generated per turn but not appended to history. For Claude and Gemini, they stayed below 25% of the output limit; we did not capture traces for o3 and Qwen.
> ### Failure modes
> *Suboptimal exploration.* Strategic exploration matters more than raw diversity. Final-turn normalized perplexity correlates positively with task score ($r = 0.324$, $p = 1.06 \times 10^{-7}$), but per-agent correlations are weak (all $|r| < 0.24, p > 0.13$), reinforcing that *what* an agent explores matters beyond *how much*.
>
> *Incomplete state-space coverage.* We introduce *on-clause coverage*: the fraction of an environment's conditional rules triggered during exploration. Per-model correlations with test-phase score reach p < 0.05 for four of five models (Qwen: r=0.392; Flash: r=0.356; o3: r=0.233; Claude: r=0.230; Gemini Pro: r=0.136 n.s.). Coverage complements perplexity: perplexity measures action *entropy*, coverage measures *yield*, whether actions exercised the environment's causal structure.
>
> *Belief rigidity.* When test-phase observations contradict exploration-phase inferences, models persist with their original interpretation. For example, a Gemini-2.5-Pro agent forms a "gravity" hypothesis from repeated noop observations, then presses right and observes a rightward shift—but attributes it to a subsequent click, preserving its gravity-only model. This is a reasoning failure, not a memory limitation (full trace in Appendix E.7).
> ## R3. Whether explicitly including query types in the prompts would encourage better exploration strategies.
> WorldTest discloses the task type but not the task parameters, so agents know *what kind* of question is coming but not *which specific question*. After normalizing action counts within each environment, human participants explore more for masked frame prediction ($1.16\times$ mean) and change detection ($1.05\times$) than for planning ($0.78\times$). No model shows comparable adaptation; exploration is query-agnostic across all five models.
> ## R4. How success rate scales with context length and whether there is saturation or diminishing returns.
> Performance does not scale with context length. Most models operate in a narrow token band (Claude 1–6%, Gemini Pro 1–16%), leaving insufficient within-model variance to test scaling. The two models with wider ranges show null (o3: $\rho = 0.01, p = 0.94$) and weakly positive (Qwen: $\rho = 0.20, p = 0.03$) partial correlations after controlling for environment difficulty.
>
> Scaling compute does not close the gap. Ranking models by cost, the 43 environments split into two regimes: **Set A** (25 envs), where performance improves with cost, and **Set B** (18 envs), where it plateaus or decreases. Set B clusters at difficulty extremes; cost scaling helps in only 33–37% of environments per task type.
>
> Neither longer context nor costlier models close the human–LLM gap, consistent with the prediction of Wang et al.[1] that purpose-trained models should improve with context but zero-shot LLMs lack the training signal to exploit it.
> ## Reference
> [1] Fan Wang et al. Context and Diversity Matter: The Emergence of In-Context Learning in World Models. In *ICLR*, 2026.

---

> > ### Author Rebuttal · Reviewer_UdVH · 2026-04-01
> >
> > Thank you for the replies. My concerns have been addressed meaningfully.

---

> > > ### Author Response · Authors · 2026-04-07
> > >
> > > Thank you for taking another careful look at our paper. We appreciate your follow-up and are glad that the additional clarifications and experiments helped address your concerns. Your comments helped us strengthen the position of the paper, and we have reflected these improvements in the revised manuscript.

---

### Official Review · Reviewer_fKo5 · 2026-03-04

**Soundness:** 2
**Presentation:** 3
**Significance:** 2
**Originality:** 3
**Overall Recommendation:** 5
**Confidence:** 3

**Summary:**

The paper introduces WorldTest, a theoretical framework for evaluating the ability of an agent to learn a world model from reward-less interactions, and the ability to use that knowledge effectively to accomplish a goal in the environment. They instantiate this with AutumnBench, a grid environment. They evaluate several models and humans on AutumnBench and analyze the differences in performance and behavior, along with limitations in existing language models.

**Compliance With Llm Reviewing Policy:**

Affirmed.

**Final Justification:**

The authors addressed my main concerns, so I increased to a 5 (accept).

**Key Questions For Authors:**

Please see the limitations section. I've been quite detailed on exactly what I'd like to see there.

**Limitations:**

- The language models used are a bit stale and no longer the frontier, and even as of the contemporaneous period before the ICML deadline models such as Opus 4.5 which leapfrogged the cited models existed. This is understandable given the long timelines in research and fast advancement pace of frontier models, but I am interested in seeing the performance of even just one modern frontier model (ones that existed prior to the contemporaneous work period are fine, such as Opus 4.5 and Sonnet 4.5). If cost is prohibitive, a cheaper model such as Sonnet 4.5 would suffice. I would prefer Opus 4.6, and a score reported with just this model would fix this limitation. There is a chance that if models such as these are not tested, the benchmark and environments could be accepted while already being saturated and irrelevant.
- Intuitively, it makes sense that if a model is easily able to infer a world model from few interactions, it should be able to complete more complex reasoning and planning tasks. However, I would want to see some empirical evidence to support this. To be convinced, I would want to see how scores in AutumnBench correlate to other benchmark scores if the goal is to test reasoning and world modeling capabilities. In robotics, works such as SIMPLER [1] run real-world correlation studies to show their benchmarks effectively transfer. In a work such as this, I would want to see such a test to validate the paper. The ideal here for me would be an agentic coding benchmark as the models need to quickly learn the environment dynamics (codebase structure, how a change affects other changes, available terminal tools, etc.) such as Terminal Bench [2] or SWE-Bench [3]. Another set of tasks could be from mathematics benchmarks. Given the limited time, I would be happy to accept numbers taken from online leaderboards with proper attribution (actually, this is preferable as it isolates the downstream evaluations from being modified in an unfaithful manner). Given that the paper has human baselines, if the correlation is close, this may serve as a cost-effective proxy to show how good LLM world modeling and reasoning abilities are relative to humans (Fig. 3) along with the differences in behavior (Fig. 5).

My low score is tentative, but I am unlikely to increase without these results. I think the idea and human baseline are good, however I am concerned about the execution and relevance with recent models. If a modern "frontier" model result (I gave plenty of options that could convince me) could be added to show that the evaluation has not saturated, and some correlation experiment with two or more reasoning and world modeling-heavy tasks such as in agentic code (Terminal Bench 2? SWE-Bench?, reasoning given in the limitations section, citing numbers with attribution is fine) to show that the grid world environment performance correlates to real tasks is added, I believe the paper would be massively improved and would be happy to increase my score substantially (not just 1 point).


[1] Xuanlin Li, et al, "Evaluating Real-World Robot Manipulation Policies in Simulation," 2024.

[2] Mike A. Merrill et al, "Terminal-Bench: Benchmarking Agents on Hard, Realistic Tasks in Command Line Interfaces," 2026.

[3] Carlos E Jimenez, et al, "SWE-bench: Can Language Models Resolve Real-world Github Issues?," in The Twelfth International Conference on Learning Representations, 2024.

**Strengths And Weaknesses:**

Strengths:
- The design of AutumnBench seems sound.
- Having human baselines is fantastic, as it compares the world modeling ability of LLMs and humans effectively. See limitations section for further thoughts on how this can be more effectively executed. I think this is the biggest strength of the paper.
- The auxiliary analysis such as usage patterns show interesting differences between human behavior and language model behaviors, and I can imagine possible avenues for future work that the community can build on.

Weaknesses:
- The paper makes claims about reasoning abilities and world model learning, but I am not convinced that observed results in a grid world correlate to performance in external benchmarks that require these abilities.
- The models used to evaluate are no longer frontier and were not state of the art as of the contemporaneous work period of 2-3 months before the deadline. I am concerned about benchmark saturation.

Please see the limitations for more details. I think this is a good but incomplete paper, which can be fixed within the rebuttal period.

---

> ### Author Rebuttal · Authors · 2026-03-31
>
> ## R1. The models evaluated are no longer frontier; report results on at least one modern model, such as Opus 4.6 or Sonnet 4.5, to rule out benchmark saturation.
>
> We thank the reviewer for this suggestion, which strengthens our contribution. We evaluated Claude Opus 4.6 using the same agent setup described in the paper. We now report per-environment results in Table E.2. In summary, Claude Opus 4.6 achieves an overall score of **39.9%** (CD: 19.8%, MFP: 44.2%, Planning: 55.8%), representing the highest performance among all reasoning models evaluated. o3 obtained the previous best result at 27.5%, followed by Sonnet 4 at 25.6%. Planning emerges as the strongest dimension, with 24 of 43 environments yielding non-zero scores, followed by MFP at 19 of 43; Change Detection remains the most challenging at 11 of 43 non-zero. Opus 4.6 achieves the highest score on 18 of 43 environments and is the first model to obtain non-zero scores on two previously unsolved environments, K8MTQ and NTQ4Y. Nevertheless, 12 AutumnBench environments continue to yield zero scores across all tasks, and the human baseline of 90.2% remains substantially ahead. These results confirm that AutumnBench is not saturated, even by current frontier models.
>
> ## R2. Whether AutumnBench scores correlate with performance on established reasoning benchmarks (for example, SWE-Bench, Terminal Bench, and math benchmarks) to validate that world modeling ability transfers to broader capabilities.
>
> We thank the reviewer for suggesting this correlation study, which provides evidence for the transferability of AutumnBench. We compared AutumnBench rankings against SWE-bench Verified [1], Terminal Bench [2], GPQA Diamond [3], and AIME 2025 [4, 5] to test whether world-modelling ability, as measured by AutumnBench, predicts performance on established reasoning benchmarks. We obtained all external scores from publicly available leaderboards as of 25 March 2026; agent scaffolds are noted in the table where applicable.
>
> | Model | AutumnBench | CD | MFP | PL | SWE-bench Verified | GPQA Diamond | TerminalBench | AIME 2025 |
> |---|---|---|---|---|---|---|---|---|
> | Claude Opus 4.6 | **39.9%** | 19.8% | 44.2% | 55.8% | 75.6% *(mini-SWE-agent)* | 89.6% | 81.8% *(ForgeCode)* | 94.2% |
> | Claude Sonnet 4 | **25.6%** | 9.5% | 30.2% | 34.9% | 74.6% *(Lingxi-v1.5)* | 83.4% | — | 74.3% |
> | o3 | **27.5%** | 22.2% | 27.9% | 32.6% | 58.4% *(mini-SWE-agent)* | — | — | 88.3% |
> | Gemini 2.5 Flash | **19.9%** | 20.3% | 25.6% | 14.0% | 35.0% *(mini-SWE-agent)* | 79.0% | 17.1% *(mini-SWE-agent)* | 73.3% |
> | Gemini 2.5 Pro | **19.4%** | 11.6% | 25.6% | 20.9% | 53.6% *(mini-SWE-agent)* | 84.4% | 32.6% *(Terminus 2)* | 87.7% |
> | Qwen3-235B | **15.7%** | 7.7% | 19.5% | 20.9% | 30.2% *(SWE-Fixer)* | 77.2% | — | 91.0% |
> | **Spearman ρ (overall)** | | | | | **0.89\*** *(n=6)* | **0.70** *(n=5)* | 0.50 *(n=3)* | 0.26 *(n=6)* |
> | **Spearman ρ (CD)** | | | | | 0.26 *(n=6)* | 0.40 *(n=5)* | −0.50 *(n=3)* | −0.14 *(n=6)* |
> | **Spearman ρ (MFP)** | | | | | **0.99\*\*** *(n=6)* | **0.82** *(n=5)* | 0.87 *(n=3)* | 0.20 *(n=6)* |
> | **Spearman ρ (PL)** | | | | | **0.90\*** *(n=6)* | 0.67 *(n=5)* | **1.00** *(n=3)* | 0.55 *(n=6)* |
>
> \* p < 0.05, \*\* p < 0.001
>
> AutumnBench overall scores exhibit a strong positive correlation with SWE-bench Verified, suggesting that world-modelling capabilities predict performance on real-world software engineering tasks. Among the individual dimensions, MFP demonstrates the strongest association with SWE-bench and GPQA Diamond, while Planning correlates with both SWE-bench and TerminalBench. By contrast, AIME 2025 shows negligible correlation with AutumnBench, indicating that mathematical competition performance may draw on fundamentally different cognitive skills than world modelling. Change Detection exhibits weak or negative correlations with all external benchmarks, including AIME. We hypothesize that this divergence arises because MFP and Planning reward pattern recognition and abstract goal reasoning, whereas CD requires models to identify precisely how environment dynamics have changed, a form of counterfactual reasoning that existing benchmarks do not directly assess. Note that the small sample sizes, particularly for Terminal Bench (n = 3), limit the statistical power of some comparisons.
>
> ## References
>
> [1] Jimenez et al. SWE-bench: Can Language Models Resolve Real-World GitHub Issues? ICLR 2024.
>
> [2] Merrill et al. Terminal-Bench: Benchmarking Agents on Hard, Realistic Tasks in CLIs. 2026.
>
> [3] GPQA Diamond Benchmark. artificialanalysis.ai/evaluations/gpqa-diamond
>
> [4] AIME 2025 Benchmark. artificialanalysis.ai/evaluations/aime-2025
>
> [5] Claude Opus 4.6 AIME score. automatio.ai/models/claude-opus-4-6

---

> > ### Author Rebuttal · Reviewer_fKo5 · 2026-03-31
> >
> > The authors have resolved all of my concerns. I have changed my score from a 3 to a 5

---

> > > ### Author Response · Authors · 2026-04-07
> > >
> > > Thank you for taking another careful look at our paper and for raising your score. We appreciate your follow-up and are glad that the additional experiments helped address your concerns. Your suggestions helped us strengthen the empirical contribution, and we have reflected these improvements in the revised manuscript.

---

### Official Review · Reviewer_ftgf · 2026-03-12

**Soundness:** 3
**Presentation:** 3
**Significance:** 4
**Originality:** 3
**Overall Recommendation:** 5
**Confidence:** 4

**Summary:**

The paper proposes WorldTest, a suite of world modelling tasks in the AutumnBench framework. AutumnBench frames their novelty around being i) a POMDP ii) that is not representation based iii) and not gym-like and iv) has a modified test environment. This leads to their design of having an agent first interact with the world, and then evaluating them on a set of possible tasks that the agent must answer using their knowledge of the world. This set of tasks includes Masked Frame Prediction (MFP), Change Detection (CD), and Planning. They demonstrate that humans substantially outperform current SotA models on the WorldTest suite, and that humans are more likely to utilize reset and no-op actions to test hypotheses,

**Compliance With Llm Reviewing Policy:**

Affirmed.

**Final Justification:**

The rebuttal addressed my concerns, many of which were about presentation and fixable in a rebuttal period. I have increase my score from a weak accept to a full accept accordingly.

**Key Questions For Authors:**

1. Why is the task type disclosed before the agent interacts with the environment? Is this necessary?
2. What does stochasticity look like in AutumnBench?
3. What is the differentiation between setA and setB in figure 4

**Limitations:**

yes

**Strengths And Weaknesses:**

Strengths
---
- Novel world-model centric test bed
- Well defined and formalized evaluation framework
- Includes human baselines and analysis and demonstrates existing gap for future work in AI research

Weaknesses
---
The biggest weakness of the paper is around the framing and presentation of the novelty. Working in the world model space, I can see the value of this benchmark, but having it stated as not gym-like and not representation-based is both unclear and uncompelling. I generally favour positive differentiators rather than negative differentiators, as they provide a more concrete contribution.

I would consider "rewardless training" or "exploration-only training" over "not gym-like".
"not representation-based" is even more ambiguous to me, and I am not convinced that WorldTest is "not representation-based".
From the paper:
"Representation-based approaches require agents to use pre-
defined output formats—such as next-frame predictions,
programs, or causal graphs"

However, WorldTest uses masked frame prediction (very similar to next-frame prediction) as one of their evaluation tasks. I would argue all three of the tasks in WorldTest (masked frame prediction, change detection, and planning) are all some form of representation. What I believe to be the real differentiator here is either: i) the representation between training (exploration) and evaluation is different AND ii) there is a set of multiple evaluation tasks, each with their own representation. This set forces (to some extent, see below) the model to rely on learned dynamics rather than directly learning an input-output mapping.

There is also a disconnect between the method and results section. The result section is not the place to introduce new concepts and components. Yet, we only learn that AutumnBench is stochastic in the results section. The focus on no-op and reset is also only introduced in the results section. Both of these should be introduced earlier. The stochastic component of AutumnBench also has to significantly clarified. What kind of stochasticity exists? Are the agents action stochastic? Are there objects in the environment that operate stochastically? Is it just the start state that is randomly determined, or is it part of the transition function?

Figure 4 is also bizarre to me. The differentiator between the two sets are critical to draw any conclusion here, but this differentiation is left to the appendix. Further, when I went to the appendix D, I could not find the differentiation listed. Where is it?

In section 5.2.2 the authors state: "Figure 6c
shows humans reset more often than models, which reset
less often and with greater variance." But if I look at figure 6c, it looks like gemini actually resets more than humans. Figure 5 shows that humans use reset as a large percentage of the action pool, but the statement itself appears incorrect and needs to be updated.

Lastly, I feel like there is a disconnect between the state intention and the final benchmark. If an agent is supposed to explore and learn robust dynamics of the world through exploration such that it can then be " (1) inferring
what will happen behind an occlusion, (2) detecting changes
in the environment’s dynamics, and (3) determining if a state
is reachable from another."
Then why is it necessary to disclose the task type before the agent interacts with the environment? If the agent is able to learn robust dynamics, it should be able to tackle any of the three evaluation tasks. Disclosing the task type before hand (even if the parameters are not disclosed) seems to provide unnecessary context.

---

> ### Author Rebuttal · Authors · 2026-03-31
>
> ## R1. Clarifying the terminology used to categorize different world model benchmarks.
> Thank you; we adopted "rewardless training" and "representation-agnostic," updating Section 1 and Table 1 accordingly. The former captures that the interaction phase provides no external reward signal. By representation-agnostic we mean that the score depends on the agent's behavior, not on its output format. Representation-based benchmarks entangle two separate questions: whether the agent understands the dynamics, and whether it can express that understanding in a required format.
>
> WorldTest uses task-specific representations to pose queries that target different aspects of environment understanding, not to directly evaluate: In MFP, the agent selects the correct frame rather than generating one; in CD, it signals the change timestep through an action; in Planning, the score reflects whether it reaches the target state. We added this clarification to Section 2.
> ## R2. Introducing stochasticity and special actions earlier in the paper rather than in the results section and clarifying the type of stochasticity (transition dynamics vs. start state vs. action noise).
> The current draft introduces stochasticity in Section 3 (via Appendix A), Section 4.2, and `no-op` and `reset` in Section 5.1. We revised Section 4.2 to define these properties earlier. It now states explicitly that stochasticity is a property of the transition function: the same action in the same state can produce different next states, implemented in Autumn through library functions such as `uniformChoice` and `randomPositions`, and introduces `no-op` and `reset` alongside it.
> ## R3. Correcting the cross-reference for the SetA/SetB differentiation in Figure 4.
> Thank you; we corrected the reference to Appendix E.3.3. We summarize here for completeness: the goal is to determine whether some environments fail to improve with additional resources, indicating limitations that scaling alone cannot overcome.
>
> Ranking models by ascending cost-per-problem (via OpenRouter)—`gemini-flash-2.5`, `qwen3-235b-a22b-thinking-2507`, `gemini-2.5-pro`, `claude-4-sonnet`, `o3`—we partition the 43 environments into SetA (25 environments, monotonic improvement with budget) and SetB (18 environments, no improvement despite additional resources).
>
> Environments in SetB are either very easy (solvable with few random actions; see Table E.6) or very hard, requiring long-horizon reasoning about stochastic dynamics. That 42% of environments do not benefit from additional compute suggests fundamental reasoning limitations beyond scaling. We added this summary alongside Figure 4.
> ## R4. Correcting the statement about human vs. model reset behavior in Section 5.2.2.
> Thank you for catching this error. We now separate the two observations and add them to the figure descriptions:
> * Figure 5: Humans allocate 12.5% of their unique actions to resets, while all reasoning models use fewer, ranging from 6.8% for Qwen down to 2.1% for Claude.
> * Figure 6c: Humans reset at least once in all 43 environments and reset more than the per-environment model mean in 33 of 43 environments. Models frequently skip resets entirely: Claude in 31 of 43 environments, Qwen in 8, Gemini Pro in 7, Gemini Flash in 4, and o3 in 3.
> ## R5. Justifying the disclosure of task type before environment interaction.
> Thank you for raising this design question. WorldTest discloses the task type (e.g., masked frame prediction) but not task parameters (e.g., which frames are masked), isolating world-model learning from task-type inference: agents know the task structure without constraining exploration strategy or internal representation.
>
> Withholding task type entirely corresponds to reward-free exploration [1], where tasks are specified only at evaluation. The reward-free literature shows this generality comes at a steep cost: task-agnostic exploration requires significantly more data for comparable performance [2], and even lightweight priors dramatically improve world-model quality [3]. AutumnBench provides this lightweight prior: testing whether agents can learn dynamics for a known task type across three distinct tasks. Even in this easier setting, the human-model gap remains large, confirming that learning the world model from interaction remains unsolved.
>
> WorldTest formally supports the fully task-agnostic setting by setting $\tau$ accordingly. Practically, task disclosure let each participant complete multiple environments per session; without it, post-interaction tutorials would make the 517-participant, 43-environment study infeasible at the scale needed for reliable human baselines. We leave ablating disclosure levels to future work (Section 6).
> ## References
> [1] Jin et al., Reward-Free Exploration for Reinforcement Learning, ICML, 2020.
>
> [2] Zhang et al. Task-Agnostic Exploration in Reinforcement Learning. NeurIPS, 2020.
>
> [3] Sekar et al. Planning to Explore via Self-Supervised World Models. ICML, 2020.

---

> > ### Author Rebuttal · Reviewer_ftgf · 2026-04-02
> >
> > I appreciate how the authors have addressed and incorporated the suggested changes and have adjusted my score accordingly.
> >
> > Further qualitative analysis of the environments that do not benefit from increased cost would be valuable toward identifying salient directions of future work, but this is a minor point given the already extensive appendices.

---

> > > ### Author Response · Authors · 2026-04-07
> > >
> > > Thank you for taking another careful look at our paper and for raising your score. We appreciate your follow-up and are glad that the additional clarifications helped address your concerns. We agree that qualitative analysis of environments that do not benefit from increased cost is a valuable direction; we have noted this as future work, including additional analysis with human and AI data. Your suggestions on terminology and presentation helped us sharpen how we frame and position the paper, and we have reflected these improvements in the revised manuscript.

---

### Official Review · Reviewer_uRQw · 2026-03-17

**Soundness:** 3
**Presentation:** 2
**Significance:** 3
**Originality:** 4
**Overall Recommendation:** 5
**Confidence:** 4

**Summary:**

This paper argues that existing ways of evaluating world models don't really test whether an agent has learned a general understanding of how an environment works. To address this, the authors propose WorldTest, a two-phase evaluation protocol where agents first freely explore an environment and then get tested on "environment-level queries" that probe global or counterfactual properties of that environment. They build a concrete benchmark called AutumnBench on top of this framework, consisting of 43 grid-world environments paired with 129 tasks spanning three query types: masked frame prediction, change detection, and planning. The experiments compare five frontier reasoning models against 517 human participants and find that humans outperform all models across the board.

**Compliance With Llm Reviewing Policy:**

Affirmed.

**Key Questions For Authors:**

1) Table E.1 shows that LLM rankings shift across prompt formulations. If someone wanted to use AutumnBench to compare two new models, how should they handle this sensitivity? Do you have recommendations for standardizing this, or do you see this as an inherent limitation?

2) You hypothesize that reset usage is linked to better world model learning. Have you tested this more directly?

**Limitations:**

yes

**Strengths And Weaknesses:**

**Warning**

I noticed that this submission violated the format requirements of ICML. The first page has clearly reduced the margin at the top of the second column and claimed more space than the limit allows. I want to raise this point to the AC, the rest of my review will be agnostic to this point.

Strengths.
The core idea of this work about evaluating world models through environment-level queries is useful and well motivated. I also found  interesting that the paper makes a clean distinction between testing what an agent knows about an environment versus how well it performs a specific task, which is something the field has been sloppy about. Authors also separate the evaluation framework from the specific benchmark instantiation. The experimental setup is solid, including  517 human participants when contrasting models vs humans.

Weaknesses

My first concern going through the work was  the generalizability of the findings this benchmark can yield. While grid worlds are a reasonable starting point, the paper's framing sometimes suggests broader applicability than what's demonstrated. The claim about benchmarking "world model learning" reads as though this covers the concept generally, when really it covers a specific slice of it. Toning down some claims or being more explicit about scope would help.

The simulator agent comparison to Dreamer-v3 (Appendix E) doesn't quite hold up. Dreamer learns a dynamics model from observations, while the simulator agent has ground truth access to transitions. These are fundamentally different, and saying they're comparable "in spirit" needs more justification or should be walked back.

Some of the behavioral analysis is correlational without clear causal links. For example, humans reset more and also perform better, but the paper frames resets as a possible explanation rather than just a covariate.

Overall, though I believe this is a good foundational work towards benchmarking environment understanding in deep learning models.

---

> ### Author Rebuttal · Authors · 2026-03-31
>
> ## Formatting concern
> Thank you for flagging this. We verified that we did not modify the ICML style files and identified that a footnote on the first page was silently dropped during compilation, causing the text to shift into the space it occupied.
> We acknowledge that this should have been caught during our pre-submission checks, and we have corrected this in the current version with the footnote properly restored and full margin compliance.
> ## R1. Whether findings from a grid-world benchmark generalize to richer domains
> We agree that the abstract should make the grid-world scope of the benchmark explicit and have revised it accordingly:
> > AutumnBench provides a framework for evaluating world-model learning in grid-world environments with environment-level queries.
>
> Sections 1 and 6 both state that AutumnBench targets grid-world environments specifically, while WorldTest is a broader framework applicable to diverse domains.
>
> Our goal with AutumnBench is to evaluate whether agents learn *underlying dynamics*. The complexity in AutumnBench lies in the dynamics encoded by the Autumn DSL, such as stochastic transitions, multi-object interactions, etc.
>
> The WorldTest protocol is domain-general in structure, though specific query types are domain-specific. To illustrate instantiation in visually complex domains, Section 6 now includes examples such as testing dynamics understanding by removing a tool and asking the agent to re-accomplish a goal, or presenting trajectories from modified environments to assess whether dynamics changed. Planning queries transfer more directly. Our primary contribution is the protocol and the empirical finding that models fail to learn dynamics even under idealized perceptual conditions.
> ## R2. Clarifying the relationship between the simulator agent and Dreamer-v3.
> Thank you for identifying this. We have revised the phrasing accordingly in Appendix E:
> > For MFP and CD, the simulator agent acts as the upper bound on what any next-state prediction approach, including learned models such as Dreamer-v3, could achieve on AutumnBench. For planning, the agent uses BFS, which is not necessarily optimal; a learned policy could plausibly outperform it, so the simulator agent serves only as a reference point for this task type.
> ## R3. Explaining the structural differences between agents and humans resets.
> We agree that this hypothesis is more of a correlation rather than a causal relation.
>
> The hypothesis came up from reviewing interaction logs (to be released upon acceptance to preserve anonymity): we observed that humans reset and re-execute similar action sequences, as though replaying trajectories to test hypotheses. By contrast, LLM scratchpad analysis (Appendix E.7) reveals that models rarely revise hypotheses after unexpected observations.
>
> To characterize agents and human reset behavior quantitatively, we designed the following analysis: we computed the **longest common subsequence (LCS) ratio** between the action sequences immediately before and after each reset. An agent that replays a similar sequence of actions after resetting would exhibit an LCS ratio near 1; unstructured resets would yield an LCS ratio near 0.
>
> Human resets show a mean LCS ratio of **0.827 (median 0.900, 95% CI [0.820, 0.835])**, far above every reasoning model: o3 0.188 (0.200), Gemini-2.5-Flash 0.273 (0.200), Qwen3-235B 0.325 (0.300), Sonnet-4 0.347 (0.317), Gemini-2.5-Pro 0.359 (0.333). We added this full analysis along with the statistical testing results in Appendix E.6., which confirms that the human–model difference is significant for each model.
>
> These structured resets in humans match the active hypothesis-testing behavior documented in prior work [1, 2]. Disentangling whether structured resets reflect hypothesis testing would require interventional designs that manipulate reset availability.
> ## R4. Handling prompt sensitivity across formulations when comparing new models.
> While prompt sensitivity is a known issue [3, 4], the human–model gap in AutumnBench remains large and stable across all prompt variants.
>
> For practitioners comparing models on AutumnBench, we recommend (1) standardizing the optimized prompt as the default in the benchmark release, serving as the primary evaluation condition; and (2) reporting results across multiple prompt formulations and runs with variance bars, treating prompt formulation as a nuisance variable. We have added a discussion of this point in the paper in Section 4.2.
>
> **References**
>
> [1] Coenen et al. (2015). Strategies to Intervene on Causal Systems Are Adaptively Selected. Cognitive Psychology, 79, 102–133.
>
> [2] Bramley et al. (2018). Intuitive Experimentation in the Physical World. Cognitive Psychology, 105, 9–38.
>
> [3] Sclar et al. (2024). Quantifying Language Models' Sensitivity to Spurious Features in Prompt Design. ICLR 2024.
>
> [4] Alzahrani et al. (2024). When Benchmarks Are Targets: Revealing the Sensitivity of Large Language Model Leaderboards. ACL 2024.

---

> > ### Author Rebuttal · Reviewer_uRQw · 2026-04-01
> >
> > I thank the authors for their detailed response. I reiterate my acceptance recommendation.

---

> > > ### Author Response · Authors · 2026-04-07
> > >
> > > Thank you for taking another careful look at our paper. We appreciate your follow-up and are glad that the additional clarifications and experiments helped address your concerns. Your comments on generalizability helped us improve how we present the paper, and we have reflected these improvements in the revised manuscript.

---

### Decision · Program_Chairs · 2026-04-30

**Decision:**

Accept (regular)

**Comment:**

Reviewers agreed that the authors propose a novel world-model-centric testbed and that the paper presents a well-defined and formalized evaluation framework. Most of the reviewers' comments were addressed during the discussion with the authors. All reviewers mentioned that the authors have resolved all of their concerns. I believe the work deserves acceptance at the conference.